# Species-specific chromatin landscape determines how transposable elements shape genome evolution

Yuheng Huang, Harsh Shukla, Yuh Chwen G Lee*

Department of Ecology and Evolutionary Biology, University of California, Irvine, Irvine, United States

**Abstract** Transposable elements (TEs) are selfish genetic parasites that increase their copy number at the expense of host fitness. The 'success', or genome-wide abundance, of TEs differs widely between species. Deciphering the causes for this large variety in TE abundance has remained a central question in evolutionary genomics. We previously proposed that species-specific TE abundance could be driven by the inadvertent consequences of host-direct epigenetic silencing of TEs—the spreading of repressive epigenetic marks from silenced TEs into adjacent sequences. Here, we compared this TE-mediated local enrichment of repressive marks, or 'the epigenetic effect of TEs', in six species in the *Drosophila melanogaster* subgroup to dissect step-by-step the role of such effect in determining genomic TE abundance. We found that TE-mediated local enrichment of repressive marks is prevalent and substantially varies across and even within species. While this TE-mediated effect alters the epigenetic states of adjacent genes, we surprisingly discovered that the transcription of neighboring genes could reciprocally impact this spreading. Importantly, our multi-species analysis provides the power and appropriate phylogenetic resolution to connect species-specific host chromatin regulation, TE-mediated epigenetic effects, the strength of natural selection against TEs, and genomic TE abundance unique to individual species. Our findings point toward the importance of host chromatin landscapes in shaping genome evolution through the epigenetic effects of a selfish genetic parasite.

*For correspondence:
grylee@uci.edu

Competing interest: The authors declare that no competing interests exist.

## Editor's evaluation

Transposable elements are genomic parasites and the fraction of the genome that is made up of such elements varies greatly between species, and models suggest that this must reflect the balance between the rate at which they multiply, and the rate at which selection purges them from the genome. Precisely how selection acts against transposable element insertions is not clear. This paper provides evidence that the strength of selection depends on the extent to which epigenetic silencing spreads to nearby genes – although the mechanism is obscure, as gene expression is not affected. This is a very interesting hypothesis that deserves more attention, and the paper is an excellent example of trying to combine population genetics models with a mechanistic understanding of the process modeled.

## Introduction

Transposable elements (TEs) are widespread genetic parasites that copy and insert themselves across host genomes. The presence and movement of TEs could impair host genome functions. TEs disrupt genes and functional elements (*Finnegan, 1992*), introduce ectopic regulatory sequences (*Chuong et al., 2017*), and trigger highly deleterious chromosomal rearrangements through nonhomologous

**eLife digest** All the instructions required for life are encoded in the set of DNA present in a cell. It therefore seems natural to think that every bit of this genetic information should serve the organism. And yet most species carry parasitic 'transposable' sequences, or transposons, whose only purpose is to multiply and insert themselves at other positions in the genome.

It is possible for cells to suppress these selfish elements. Chemical marks can be deposited onto the DNA to temporarily 'silence' transposons and prevent them from being able to move and replicate. However, this sometimes comes at a cost: the repressive chemical modifications can spread to nearby genes that are essential for the organism and perturb their function.

Strangely, the prevalence of transposons varies widely across the tree of life. These sequences form the majority of the genome of certain species – in fact, they represent about half of the human genetic information. But their abundance is much lower in other organisms, forming a measly 6% of the genome of puffer fish for instance. Even amongst fruit fly species, the prevalence of transposable elements can range between 2% and 25%. What explains such differences?

Huang et al. set out to examine this question through the lens of transposon silencing, systematically comparing how this process impacts nearby regions in six species of fruit flies. This revealed variations in the strength of the side effects associated with transposon silencing, resulting in different levels of perturbation on neighbouring genes. A stronger impact was associated with the species having fewer transposons in its genome, suggesting that an evolutionary pressure is at work to keep the abundance of transposons at a low level in these species. Further analyses showed that the genes which determine how silencing marks are distributed may also be responsible for the variations in the impact of transposon silencing. They could therefore be the ones driving differences in the abundance of transposons between species.

Overall, this work sheds light on the complex mechanisms shaping the evolution of genomes, and it may help to better understand how transposons are linked to processes such as aging and cancer.

recombination (*Langley et al., 1988*; *Montgomery et al., 1991*). Nevertheless, the ability to self-replicate has allowed TEs to successfully occupy nearly all eukaryotic genomes surveyed (reviewed in *Wells and Feschotte, 2020*). Within a eukaryotic genome, TEs are prevalent in both gene-poor, repeat-rich heterochromatic and gene-rich euchromatic regions (e.g., *Kaminker et al., 2002*; *Bergman et al., 2006*). TEs in the heterochromatic genome are oftentimes fragmented, losing their ability to replicate (e.g., *Hoskins et al., 2007*; *Hoskins et al., 2015*). On the contrary, euchromatic TEs, which could intersperse with functional elements, commonly retain the potential to replicate, making them active players for not only their own evolutionary dynamics, but also the function and evolution of the euchromatic genome. Because of that and the technical challenges associated with identifying TEs in the heterochromatic regions (*Salzberg and Yorke, 2005*; *Treangen and Salzberg, 2011*), studies have been largely focused on the evolution of TEs in the euchromatic genome. Intriguingly, the abundance of TEs in the euchromatic genome substantially varies across the phylogenetic tree (*Huang et al., 2012*; *Elliott and Gregory, 2015*; *Wells and Feschotte, 2020*). For instance, in the assembled vertebrate genomes, which mostly consist of euchromatic regions, the proportion occupied by TEs ranges from only 6% in pufferfish (*Volff et al., 2003*) to more than 65% in salamander (*Nowoshilow et al., 2018*). Even within the same genus, genomic TE abundance differs widely (e.g., 2.5–25% of assembled genome sequences in *Drosophila*; *Clark et al., 2007*; *Rius et al., 2016*). Deciphering the role of this prevalent parasite in shaping genome evolution has remained a central question in genomics (*Kazazian, 2004*; *Feschotte and Pritham, 2007*; *Arkhipova and Kumar, 2018*); however, the ultimate causes of such dramatic divergence in TE abundance in the euchromatic genome remain unclear.

Theoretical analyses proposed that, in panmictic host populations with unrestricted recombination, TE abundance is determined by how quickly TEs replicate and how fast they are removed from the populations by natural selection against their harmful fitness effects (*Charlesworth and Charlesworth, 1983*, reviewed in *Lee and Langley, 2010*). Under this model, divergent genome-wide TE abundance could be driven by between-species differences in the strength of selection against TEs. Currently available evolutionary models that address this possibility have focused on population

genetic parameters that influence the *efficacy of selection* removing TEs, such as mating systems (*Wright and Schoen, 1999*; *Dolgin and Charlesworth, 2006*; *Boutin et al., 2012*) and effective population size (*Lynch and Conery, 2003*). Yet, empirical support for such a hypothesis has been mixed (*Dolgin et al., 2008*; *Lockton and Gaut, 2010*; *de la Chaux et al., 2012*; *Arunkumar et al., 2015*; *Agren et al., 2014*; *Mérel et al., 2021*; *Oggenfuss et al., 2021*). On the other hand, between-species differences in the magnitude of harmful effects exerted by TEs, and accordingly the strength of selection against TEs, could also determine genomic TE abundance, a plausible hypothesis that is yet to have empirical investigations.

A new avenue for exploring how these genetic parasites shape the function and evolution of eukaryotic genomes was opened by the recently discovered host-directed silencing of TEs and the associated 'inadvertent' deleterious epigenetic effects (reviewed in *Choi and Lee, 2020*). To counteract the selfish increase of TEs in host genomes, eukaryotic hosts have evolved small RNA-mediated mechanisms to transcriptionally silence TEs (reviewed in *Slotkin and Martienssen, 2007*; *Czech et al., 2018*; *Deniz et al., 2019*). Host protein complexes are guided by small RNAs to TEs with complementary sequences, which is followed by the recruitment of methyltransferases that modify DNA or histone tails at TE sequences (*Qi et al., 2006*; *Aravin et al., 2008*; *Wang and Elgin, 2011*; *Sienski et al., 2012*; *Le Thomas et al., 2013*). Such a process results in the enrichment of DNA methylation or di- and tri-methylation on lysine 9 of histone H3 (H3K9me2/3), both repressive epigenetic modifications that are typically found in heterochromatic regions and associated with repressed gene expression (reviewed in *Pikaard and Mittelsten Scheid, 2014*; *Allis and Jenuwein, 2016*). This repressed transcription of TEs results in reduced RNA intermediates (for RNA-based TEs) and proteins (e.g., transposase and reverse transcriptase) necessary for TE replication, effectively slowing the selfish propagation of TEs.

While such epigenetic silencing of TEs should benefit their hosts, studies in various model species have found that repressive marks enriched at silenced TEs 'spread' beyond TE boundaries, leading to local enrichment of such marks at TE-adjacent sequences across the euchromatic genomes (i.e., *Mus*, *Drosophila, Arabidopsis,* and *Oryza*; *Rebollo et al., 2012*; *Sienski et al., 2012*; *Pezic et al., 2014*; *Lee, 2015*; *Quadrana et al., 2016*; *Choi and Purugganan, 2018*, reviewed in *Choi and Lee, 2020*). Furthermore, TEs with such effects were observed to have lower population frequencies (*Hollister and Gaut, 2009*; *Lee, 2015*; *Lee and Karpen, 2017*), suggesting that selection acts to remove them. These discoveries highlight the potential importance of TE-triggered epigenetic effects in shaping genome evolution. Interestingly, the strength of TE-mediated local enrichment of repressive marks substantially varies between distantly related taxa (reviewed in *Choi and Lee, 2020*). Investigations on pairs of closely related species further revealed that this 'epigenetic effect of TE' differs and is stronger in the species with fewer euchromatic TEs (*Arabidopsis thaliana* vs. *Arabidopsis lyrata* and *Drosophila melanogaster* vs. *Drosophila simulans*; *Hollister et al., 2011*; *Lee and Karpen, 2017*). These observations spurred our previous hypothesis that, across species, different TE-mediated enrichment of repressive marks could result in varying functional consequences and thus differences in the strength of selection against TEs, eventually contributing to divergent TE abundance in the euchromatic genome (*Lee and Karpen, 2017*). We further postulated that this difference in TE-mediated epigenetic effects could have resulted from species-specific genetic modulation of the repressive chromatin landscape (*Lee and Karpen, 2017*), which was shown to determine the spreading of repressive epigenetic marks from constitutive heterochromatin into adjacent euchromatic sequences (reviewed in *Girton and Johansen, 2008*; *Elgin and Reuter, 2013*).

To fully examine the hypothesis that varying host chromatin landscape drives between-species differences of TEs through epigenetic mechanisms, one needs to connect species-specific regulation of chromatin landscape, TE-mediated enrichment of repressive marks, the associated functional consequence and resultant selection against TEs, and genomic TE abundance. However, former analyses that compared TE-mediated epigenetic effects between species have limited sampling (two species) and thus lack sufficient statistical power for robust inference (*Hollister et al., 2011*; *Lee and Karpen, 2017*). Also, support for key links of the hypothesis is lacking. For instance, selection against TE-mediated epigenetic effects was expected to result from the associated reducing effects on the expression of neighboring genes. Yet, investigations in multiple taxa reported weak or no associations between the epigenetic effects of TEs and neighboring gene expression (*Quadrana et al., 2016*; *Stuart et al., 2016*; *Lee and Karpen, 2017*; *Choi and Purugganan, 2018*, reviewed in *Kelleher*

*et al., 2020*; *Choi and Lee, 2020*). These inconclusive observations cast doubt on the possibility that TE-mediated epigenetic effects impair host fitness by silencing neighboring genes and whether this particular deleterious consequence indeed shapes genome evolution. Furthermore, previous comparisons of population frequencies between TEs with and without epigenetic effects, an approach used to infer the strength of natural selection removing TEs, could not exclude the confounding influence of other harmful effects of TEs on their population frequencies (e.g., *Hollister and Gaut, 2009*; *Lee, 2015*). Accordingly, those analyses could not unequivocally support selection against TE-mediated epigenetic effects. Multi-species studies that span an appropriate evolutionary distance and connect the missing links in the proposed hypothesis would be needed to test the predicted importance of TE-mediated epigenetic effects in determining between-species differences of TEs.

In this study, we investigated the prevalence of TE-mediated local enrichment of repressive epigenetic marks, or 'TE-mediated epigenetic effects', in the euchromatic genome of six species in the *D. melanogaster* subgroup (diverged around 10 MYR; *Obbard et al., 2012*, *Figure 1A*). These species are from the two well-studied species complexes (*melanogaster* and *yakuba* complexes), providing good phylogenetic resolution to address the role of TE-mediated epigenetic effects in genome evolution. While TE insertions in all species studied result in robust local enrichment of repressive epigenetic marks, the strength of such effects varies substantially within genomes, among species, and between species complexes. Our larger sample size allowed us to re-examine the still debated question about the impacts of TE-mediated enrichment of repressive marks on neighboring gene expression, which surprisingly revealed their complex interactions. Importantly, our multi-species analysis provides the power to test the predicted associations between TE abundance in the euchromatic genome, TE-mediated epigenetic effects, selection against such effects, and host chromatin environment, while uncovering the evolutionary causes for the wide variety of TE abundance between species.

## Results

### TE-mediated local enrichment of repressive marks is prevalent across studied species

We investigated whether previously reported local enrichment of heterochromatic marks around euchromatic TEs in model *Drosophila* species (*Lee, 2015*; *Lee and Karpen, 2017*) is prevalent across species in the *D. melanogaster* subgroup. H3K9me2 and H3K9me3 are histone modifications that are highly enriched in the constitutive heterochromatin in *D. melanogaster* (*Riddle et al., 2011*; *Kharchenko et al., 2011*) and are generally considered 'heterochromatic marks'. Previously, it was shown that TEs lead to a local enrichment of both of these two histone modifications in the *D. melanogaster* euchromatic genome (*Lee and Karpen, 2017*), and we chose to focus on one of them (H3K9me2) in this study. We performed spike-in controlled chromatin immunoprecipitation (ChIP)-seq targeting H3K9me2 using 16–18 hr embryos (see Materials and methods). We estimated histone modification density (HMD), which is the ratio of fragment coverage in ChIP samples to that in matching input samples, standardized by the ratio of spike-in fragments (*Lam et al., 2019*) (see Materials and methods). TE insertions in strains used for the ChIP-seq experiment were annotated by running Repeatmodeler2 (*Flynn et al., 2020*) on genomes assembled using long-read PacBio sequencing. The high continuity of these genomes enables a more comprehensive identification of TEs than previous studies based on short-read sequencing data (*Khost et al., 2017*; *Chakraborty et al., 2019*). Because our analysis mainly focuses on the evolutionary dynamics of euchromatic TEs and their role in driving the evolution of the euchromatic genome, we excluded TEs in or near heterochromatic regions from our analysis (see Materials and methods).

Across all six species analyzed, we observed significant enrichment of H3K9me2 HMD around TEs, and this enrichment decreases to the background level within 10 kb (*Figure 1B*). To exclude the possibility that this enrichment of H3K9me2 is due to TE preferentially inserted into regions that are already enriched with heterochromatic marks (e.g., *Dimitri and Junakovic, 1999*), we collected H3K9me2 epigenomic data for two genomes of *D. simulans* and *Drosophila yakuba* and compared the enrichment of H3K9me2 for homologous sequences with and without a TE insertion. The H3K9me2 enrichment is observed in the vicinity of TEs in the genome where they are present, but not at homologous sequences in the other genome (*Figure 1C*). This observation expanded previous single-species

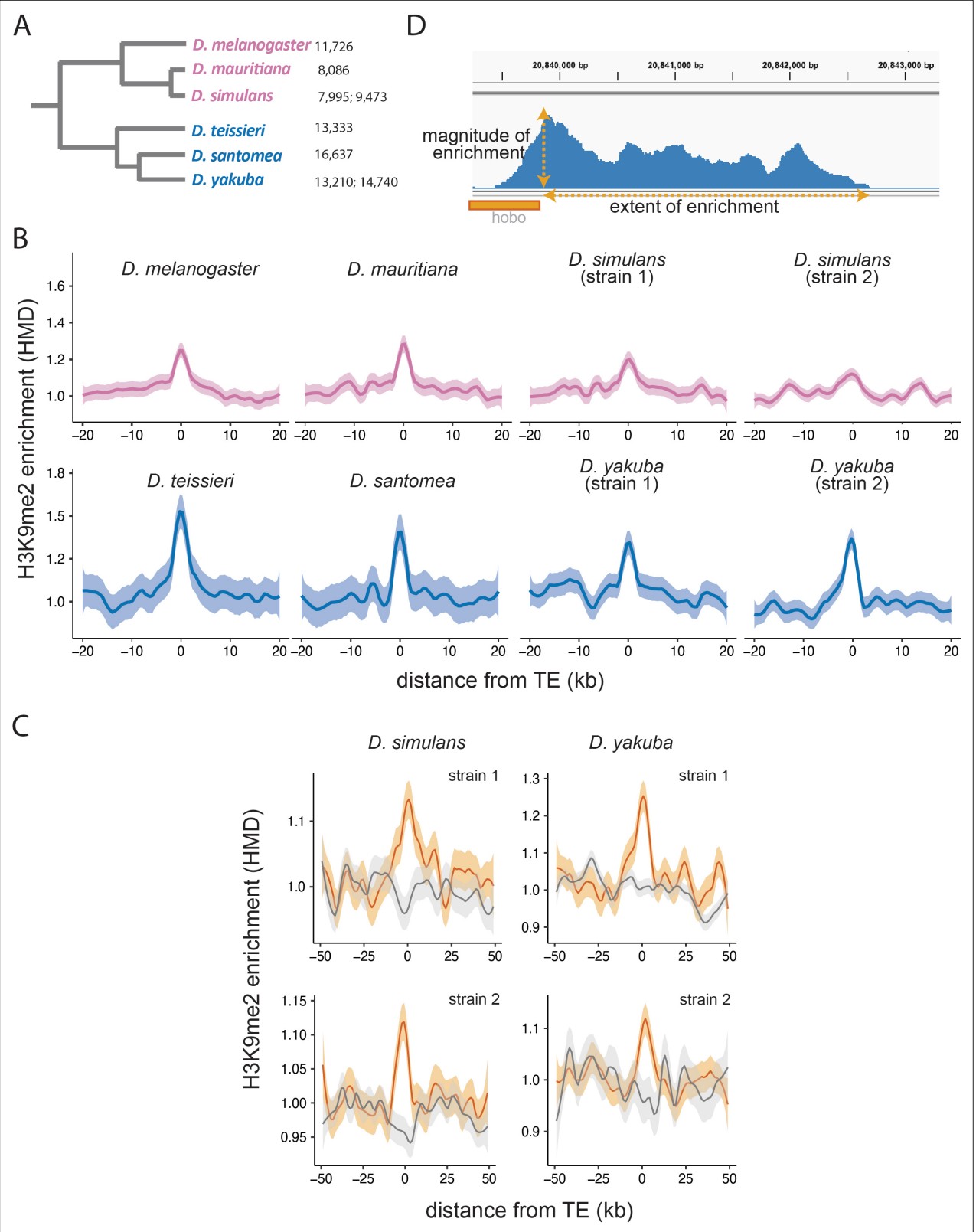

**Figure 1.** Enrichment of H3K9me2 around euchromatic transposable elements (TEs) across species. (**A**) Phylogenetic relationship among species included in this study. Species in the *Drosophila melanogaster* complex are in pink, while those in *Drosophila yakuba* complex are in blue. Numbers after each species denote the number of euchromatic TEs called by RepeatModeler2, before assignment into TE families and merging of adjacent copies (see Materials and methods). (**B**) Genome-wide average H3K9me2 HMD levels around euchromatic TEs with LOESS smoothing (span = 15%) in

*Figure 1 continued on next page*

*Figure 1 continued*

studied genomes; 95% confidence interval around smooth is shown as shaded areas. (**C**) Genome-wide average H3K9me2 HMD levels at homologous sequences in the presence (orange) and absence (gray) of euchromatic TEs in species that have data for two strains (*Drosophila simulans* and *D. yakuba*). The average H3K9me2 HMD level was smoothed with LOESS (span = 15%) with 95% confidence intervals around smooth shown as the shaded areas. (**D**) An Integrated-Genome-Viewer view showing the local enrichment of H3K9me2 around a *hobo* TE, and the two estimates (magnitude and extent of H3K9me2 enrichment) estimated for quantifying the epigenetic effects of individual TE.

The online version of this article includes the following figure supplement(s) for figure 1:

**Figure supplement 1.** Associations between estimates for the magnitude and extent of transposable element (TE)-mediated H3K9me2 enrichment that are based on one genome (x-axis) or two genomes (y-axis).

**Figure supplement 2.** Associations between the magnitude and extent of transposable element (TE)-mediated H3K9me2 enrichment.

**Figure supplement 3.** Associations between the magnitude/extent of transposable element (TE)-mediated H3K9me2 enrichment and TE length.

**Figure supplement 4.** The magnitude and extent of transposable element (TE)-mediated H3K9me2 enrichment for TEs that are full length (at least 70% of canonical sequence length) and truncated.

**Figure supplement 5.** Antibody specificity assay using SNAP-ChIP K-MetStat Panel.

**Figure supplement 6.** Associations between replicates for the magnitude and extent of transposable element (TE)-mediated H3K9me2 enrichment.

**Figure supplement 7.** The –log10 scores for the significance of called peaks in replicates are shown on the X-Y plot, or diagnostic IDR plots.

**Figure supplement 8.** Associations for the extent of transposable element (TE)-mediated H3K9me2 enrichment with different HMD cutoffs: HMD > 1 (used threshold throughout the study), HMD > 1.5, and HMD > 2.

observations (*Lee, 2015*; *Lee and Karpen, 2017*) and supported that the local enrichment of heterochromatic marks in the euchromatic regions is induced by TEs in multiple species.

## Strength of TE-mediated epigenetic effects depends jointly on TE attributes and host genetic background

In order to investigate biological factors associated with the strength of TE-mediated epigenetic effects, we quantified the 'magnitude' and 'extent' of local enrichment of H3K9me2 for *individual* TEs (*Figure 1D*). We estimated the magnitude of TE-mediated epigenetic effects as the H3K9me2 enrichment in the 1 kb window immediately adjacent to a TE insertion. To identify the extent of TE-mediated enrichment of H3K9me2, we scanned from TE to locate the farthest 1 kb window in which H3K9me2 enrichment level is above that of the local background (see Materials and methods for details). For *D. simulans* and *D. yakuba,* we also estimated the magnitude and extent of TE-induced H3K9me2 enrichment by comparing H3K9me2 enrichment at homologous sequences between strains with and without focal TEs (two-genome estimates; see Materials and methods). Estimates based on one genome or two genomes strongly correlate (*Spearman rank correlation coefficient* $\rho$ =0.64–0.85 [magnitude] and 0.40–0.64 [extent], p<$10^{-10}$ for all tests, *Figure 1—figure supplement 1*). Because an important aspect of our analyses is the comparison of TE-induced H3K9me2 across species, we reported analyses based on single-genome estimates henceforth.

The assembly of constitutive heterochromatin has been proposed to depend on the concentration of heterochromatic enzymes and structural proteins, which should be the highest at the nucleation site and gradually decrease, leading to the *cis* 'spreading' of repressive marks (*Locke et al., 1988*). Under this model, the magnitude of the enrichment for repressive marks at the nucleation site should determine the extent of the enrichment for repressive marks. Yet, predictions of this model were found to be inconsistent with several empirical observations, including the discontinuous 'spreading' of repressive marks from constitutive heterochromatin (*Belyaeva and Zhimulev, 1991*; *Talbert and Henikoff, 2000*) and the dependency of the extent of such effect on factors other than heterochromatin mass (*Sabl and Henikoff, 1996*). If similar molecular mechanisms are also applicable to epigenetically silenced TE in the euchromatic genome, the magnitude and the extent of TE-mediated local enrichment of H3K9me2 would not perfectly correlate and should capture different aspects of such TE-mediated effects. While we have no a priori predictions for the relative importance of these two indexes in the questions that this study aims to address, we anticipate that the extent of TE-mediated H3K9me2 enrichment should more likely be influenced by local genomic context than the magnitude of such effect. Nevertheless, we found that the magnitude and extent of TE-mediated H3K9me2 enrichment strongly correlated within genomes (*Spearman rank correlation coefficient* $\rho$ =0.58–0.75, p<$10^{-16}$, *Figure 1—figure supplement 2*).

TE length was postulated to be an important factor determining the strength of TE-mediated epigenetic effects because silenced TEs that are longer in length are expected to represent larger heterochromatin mass (*Lee, 2015*). Consistent with the prediction, we observed significant, though weak, positive correlations between TE length and the strength of TEs' epigenetic effects within most genomes studied (*Spearman rank correlation coefficient* $\rho$ =0.12–0.22, p<0.05; *Figure 1— figure supplement 3*). It is worth noting that previous analysis on non-reference *D. melanogaster* strains was unable to test this prediction, due to the inability to assemble internal sequences of TEs with short-read resequencing data (*Lee and Karpen, 2017*). For the same reason, previous analysis could not study whether the epigenetic effects differed between full-length and truncated TEs that have different potential to be transcribed. This distinction between TE insertions could be important because TE-mediated local enrichment of repressive marks was mainly observed with transcriptionally active TEs (*Pezic et al., 2014*, but see *Sentmanat and Elgin, 2012*). Consistently, we found that full-length TEs exert significantly larger magnitude and extent of H3K9me2 enrichment than truncated TEs in several genomes (*Mann-Whitney U test,* p<0.05 for *D. melanogaster, Drosophila mauritiana, Drosophila santomea* (magnitude) and for *D. melanogaster* and *D. santomea* (extent); *Figure 1— figure supplement 4*).

We next compared TEs of different classes, which are classifications based on the transposition mechanisms of TEs (*Wicker et al., 2007*) and previously observed to associate with varying strength of epigenetic effects within *D. melanogaster* genomes (*Lee, 2015*; *Lee and Karpen, 2017*). While different classes of TEs showed similar levels of H3K9me2 enrichment, there is a very strong species-complex effect in which TEs in genomes of *yakuba* complex showed much larger *magnitude* of epigenetic effects than those in genomes of *melanogaster* complex (*Figure 2A*, an average 1.7-fold larger). On the other hand, the *extent* of H3K9me2 enrichment is more variable and does not show a similar trend (*Figure 2A*). Interestingly, the extent and magnitude of H3K9me2 spreading vary substantially between TEs of different families within a class and between TEs of the same family (*Figure 2B* for *D. simulans* and see *Figure 2—figure supplement 1* for other *melanogaster* complex species and *Figure 2—figure supplement 2* for *yakuba* complex species). Moreover, the rank order of the extent and magnitude of TE-induced H3K9me2 enrichment of TE families varies between species and even between strains of the same species. These observations strongly suggest that the strength of TE-mediated epigenetic effects depends on both TE family attributes and host genetic background. It is worth noting that the percentage of TEs with a family assigned is higher in the *melanogaster* complex than that in the *yakuba* complex (*Figure 2—figure supplement 3*), and our analysis likely missed TE families that are highly divergent in and/or unique to the *yakuba* complex species.

To further study the effects of host genome-by-family interaction across species, we compared the epigenetic effects of TEs from families that are found in all strains and have at least five copies (*Figure 2C*). Interestingly, while some TE families show universally strong H3K9me2 enrichment (e.g., 1360 for the magnitude) or weak H3K9me2 enrichment (e.g., FB for the magnitude) across species, some families' epigenetic effects clearly depend on host genotype (e.g., H element shows larger magnitude and extent of H3K9me2 enrichment in *yakuba* complex species than in *melanogaster* complex species). Using a linear regression model, we found significant family-by-strain interaction effects for both the magnitude (ANOVA F-value: 3.4, df = 35, p=1.4 × 10$^{-10}$) and the extent (ANOVA F-value: 4.4, df = 35, p<5.8 × 10$^{-16}$) of H3K9me2 enrichment. While the significant host genome-by-family interaction on the *magnitude* of H3K9me2 enrichment could be strongly driven by Cr1a family, all TE families investigated showed strong host genome-by-family interactions for the *extent* of H3K9me2 enrichment (*Figure 2C*).

In addition to the intrinsic biological attributes of individual TEs, the insertion locations of TEs may also influence the magnitude and extent of their epigenetic effects, especially given that the chromatin environment is quite different between genic and non-genic sequences (*Filion et al., 2010*; *Kharchenko et al., 2011*). We categorized TEs according to their insertion locations relative to genes (intergenic, intronic, and exonic), but did not find a significant difference between them in terms of the magnitude of H3K9me2 enrichment except in a few cases (intergenic > exonic (*Drosophila teissieri* and *D. santomea*); exonic > intergenic ~ intronic (*D. simulans* strain 1); *Mann-Whitney U test,* p<0.05; *Figure 2—figure supplement 4*). On the other hand, for the extent of TE-mediated H3K9me2 enrichment, intergenic TEs exert larger such effects than other TEs in multiple genomes (*Mann-Whitney U test,* p<0.05 for *D. simulans* strains 1 and 2, *D. santomea,* and *D. yakuba* strain 2; *Figure 2—figure*

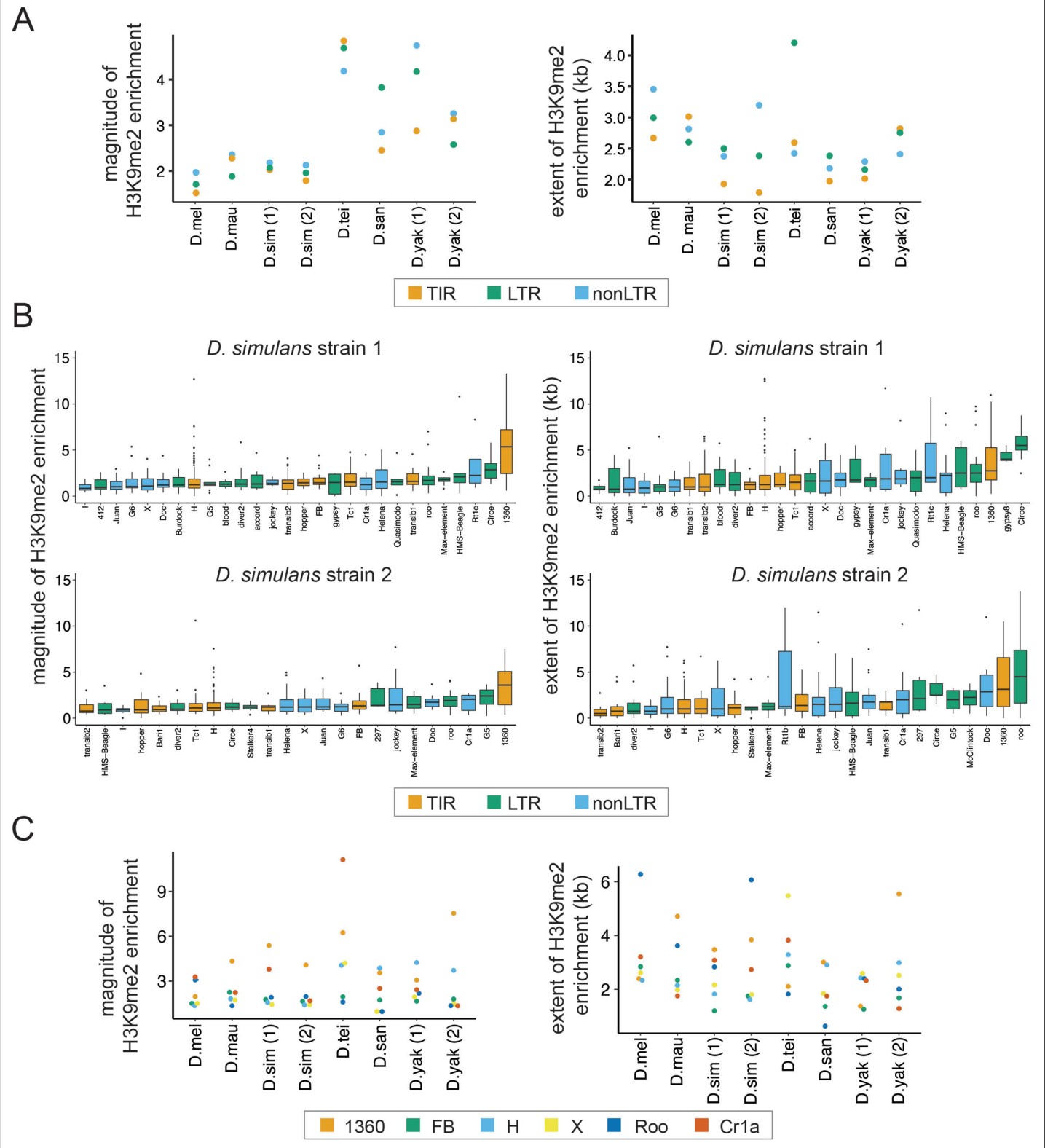

**Figure 2.** Variation in the epigenetic effects of transposable elements (TEs) within genomes. (**A**) The mean magnitude (left) and extent (right) of TE-induced H3K9me2 enrichment for different types of TEs in eight genomes from six species are shown. Different colors represent TEs of different classes, including Terminal Inverted Repeat (TIR), Long Terminal Repeat (LTR), and non-Long Terminal Repeat (non-LTR, also known as LINE) insertions. (**B**) The magnitude (left) and extent (right) of TE-induced H3K9me2 enrichment for different TE families in the two strains of *Drosophila simulans*. Only TE

*Figure 2 continued on next page*

*Figure 2 continued*

families with at least five identified copies in a genome were included. See *Figure 2—figure supplements 2 and 3* for other genomes. (**C**) The median magnitude (left) and extent (right) of TE-induced H3K9me2 enrichment for six TE families with at least five copies in all genomes studied.

The online version of this article includes the following figure supplement(s) for figure 2:

**Figure supplement 1.** The magnitude and extent of transposable element (TE)-mediated H3K9me2 enrichment of different families for other *melanogaster* complex species (*Drosophila melanogaster* and *Drosophila mauritiana*).

**Figure supplement 2.** The magnitude and extent of transposable element (TE)-mediated H3K9me2 enrichment of different families for *yakuba* complex species (*Drosophila teissieri, Drosophila santomea,* and *Drosophila yakuba*).

**Figure supplement 3.** The percentage of transposable elements (TEs) assigned to a TE family based on blast analysis.

**Figure supplement 4.** Comparisons of the magnitude and extent of transposable element (TE)-mediated H3K9me2 for intergenic, intronic, and exonic TEs.

*supplement 4*). Together, our observations revealed that the epigenetic effects of TEs *jointly* depend on the class, family identity, length, and insertion locations of TEs as well as the host genetic background (also see below).

## Complex relationship between TE-induced enrichment of H3K9me2 and neighboring gene expression

Euchromatic TEs are interspersed with actively transcribing genes, and TE-mediated spreading of H3K9me2 was previously observed to extend into neighboring genes (*Lee, 2015*; *Lee and Karpen, 2017*). The enrichment of H3K9me2, a repressive histone modification, is generally associated with suppressed gene expression (*Kouzarides, 2007*). Accordingly, TE-mediated epigenetic effects were expected to lower neighboring gene expression. Yet, previous studies in several model species found limited effects of TE-induced enrichment of repressive marks on adjacent gene expression (reviewed in *Kelleher et al., 2020*; *Choi and Lee, 2020*). These observations left an important yet unsolved question about the functional importance of TE-mediated epigenetic effects.

To revisit this question with expanded data, we studied whether TE-mediated epigenetic effects associate with H3K9me2 enrichment or the expression level of their *nearest* adjacent genes. Because TEs inside genes could alter gene expression through other mechanisms and potentially confound the analysis (e.g., the disruption of functional sequences, see *Kelleher et al., 2020*; *Choi and Lee, 2020*), we only included intergenic TEs for analyses in this section. Similar to previous observations made in *D. melanogaster* (*Lee, 2015*; *Lee and Karpen, 2017*), *H3K9me2 enrichment* level at gene body positively correlates with the magnitude and extent of TE-induced H3K9me2 enrichment in all species analyzed (*Spearman rank correlation coefficient* $\rho$ =0.16–0.33 [magnitude] and 0.18–0.31 [extent], p<0.05 for all tests, *Figure 3A*). Importantly, for the magnitude of TE-mediated epigenetic effects, this association is much stronger for genes close to TEs than those far from TEs (regression analysis: genic H3K9me2~TE epigenetic effects + close/far from genes + interaction, p-value for interaction term <0.05 for all genomes except for *D. mauritiana* [p=0.06] and *D. teissieri* [p=0.09]; *Figure 3— figure supplement 1*). For the extent of TE-mediated epigenetic effects, we observed similar, but weaker, distance-dependent effects (regression analysis, p-value for interaction term <0.001 for *D. mauritiana, D. simulans* strain 1, *D. teissieri,* and p=0.06 for *D. melanogaster*; *Figure 3—figure supplement 2*). Regression analysis without binning genes into those close or distant to TEs reached a similar conclusion that the associations between TE-mediated epigenetic effects and genic H3K9me2 enrichment depend on gene-TE distance (genic H3K9me2 ~ TE epigenetic effects + distance to TE + interaction; p-value for interaction term <0.05 for all tests, except for the extent of the effect for *D. yakuba* strain 1, p=0.05).

Surprisingly, we found nearly absent correlations between TE-mediated H3K9me2 enrichment and the *expression rank* (ranking from the highest expressed genes) of adjacent genes in all genomes, except in one incidence (*Spearman rank correlation tests*, p>0.05 for all tests except for the extent of the effect of *D. yakuba* strain 2, *Spearman rank correlation coefficient* $\rho$ =0.17, p<0.01; *Figure 3A*). Furthermore, the associations between gene expression and the epigenetic effects of TEs do not differ between close and distant gene-TE pairs for most genomes (gene expression ~ TE epigenetic effects + close/far from genes + interaction, p-value for interaction term >0.05 for all genomes except for *D. simulans* strain 1 (magnitude) and *D. santomea* (extent); *Figure 3—figure supplements 3*

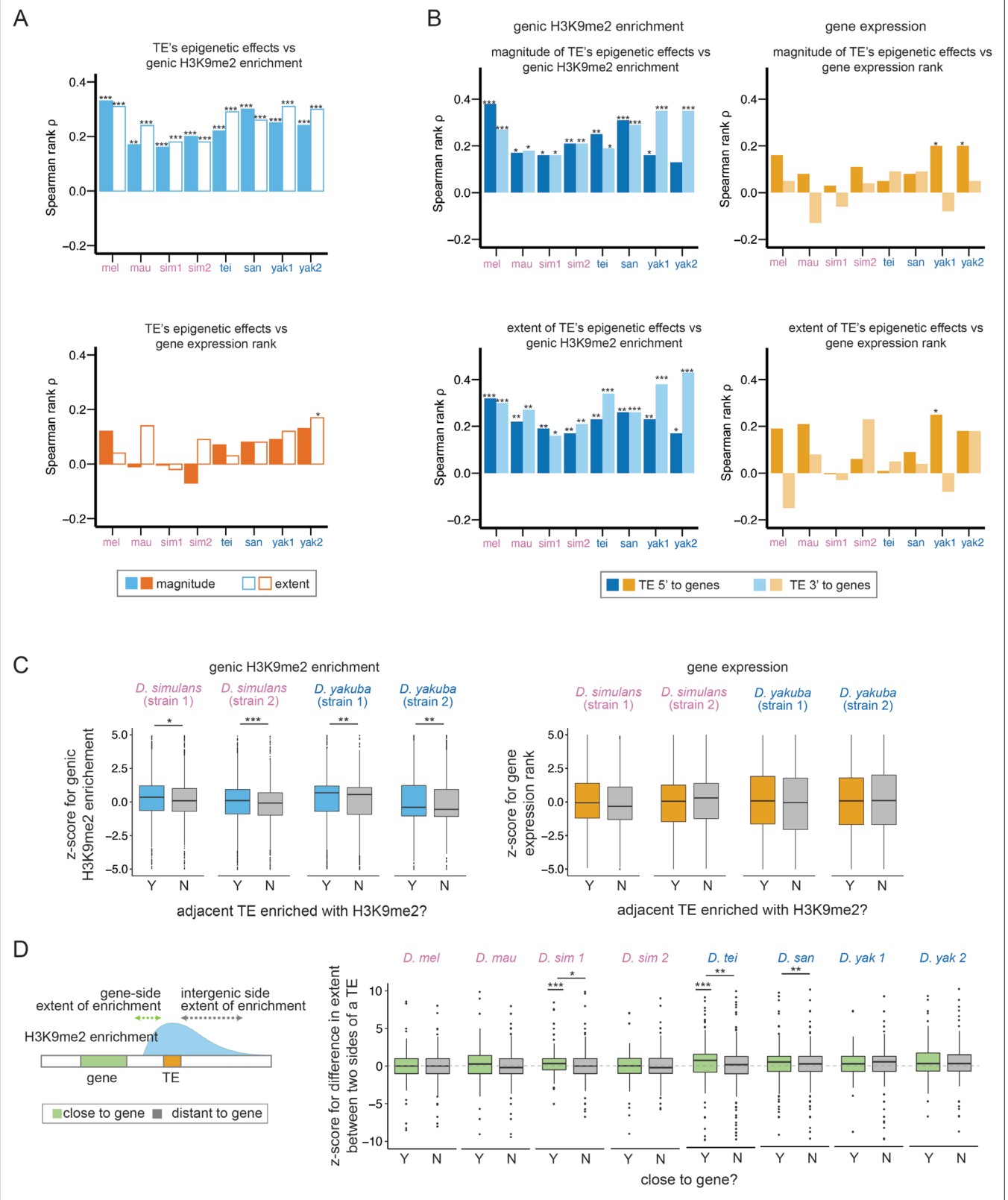

**Figure 3.** Associations between transposable element (TE)-mediated H3K9me2 enrichment and the epigenetic states and expression of neighboring genes. (**A**) *Spearman rank correlation coefficients ($\rho$) between the magnitude (filled bars) and extent (open bars) of TE-mediated H3K9me2 enrichment and genic H3K9me2 enrichment level (top) and gene expression rank (higher rank means lower expression; bottom) of nearby genes. Most of the Spearman rank correlation coefficients are significantly different from 0 for comparisons of genic H3K9me2 enrichment level (top), but not for*

*Figure 3 continued on next page*

*Figure 3 continued*

comparisons of gene expression rank (bottom). (**B**) *Spearman rank correlation coefficients ($\rho$)* between the magnitude (top) and extent (bottom) of TE-mediated H3K9me2 enrichment and the H3K9me2 enrichment level (left) and gene expression rank (right) of nearby genes for TEs 5′ (dark blue/orange) and 3′ (light blue/light orange) to genes. For the genic H3K9me2 enrichment, all *Spearman rank correlation coefficients* are significantly different from 0 except for one test. For gene expression rank, few correlations are significantly different from 0. (**C**) z-Scores for comparing the H3K9me2 enrichment (left) and expression rank (right) of homologous genic alleles whose nearby TEs with (blue/orange) or without (gray) epigenetic effects (as defined as the magnitude of H3K9me2 enrichment >1; see *Figure 3—figure supplement 7* for categorizing TEs with the extent of H3K9me2 enrichment, which gives similar results). A positive z-score means the allele with TE has higher H3K9me2 enrichment or larger expression rank (i.e., lower expression level) than the homologous allele without TE in another strain. (**D**) A cartoon describing the 'genic side' and 'intergenic side' extent of H3K9me2 enrichment mediated by TEs is shown on the left. z-Scores for comparing the extent of TE-mediated H3K9me2 enrichment on the intergenic side and on the genic side for TEs close to (green) and far (gray) from genes whose expression is at least 10 RPKM. A positive z-score means that the extent of TE-mediated H3K9me2 enrichment is more restricted on the genic side than the intergenic side. In several genomes, z-scores for TEs close to genes are significantly different from 0 and/or larger than those for TEs distant to genes. mel: *Drosophila melanogaster,* mau: *Drosophila mauritiana,* sim1: *Drosophila simulans* strain 1, sim2: *D. simulans* strain 2, tei: *Drosophila teissieri,* san: *Drosophila santomea,* yak1: *Drosophila yakuba* strain 1, yak2: *D. yakuba* strain 2. ***p<0.001, **p<0.01, *p<0.05 for *Spearman rank correlation tests* (**A, B**) and *Mann-Whitney U tests* (**C, D**).

The online version of this article includes the following figure supplement(s) for figure 3:

**Figure supplement 1.** Associations between the magnitude of transposable element (TE)-mediated H3K9me2 enrichment and genic H3K9me2 enrichment are much stronger for genes close to TEs (distance to a TE is smaller than the 50% quantile, blue) than for genes distant to TEs (gray; see *Figure 3—figure supplement 2* for the extent of TE-mediated H3K9me2 enrichment).

**Figure supplement 2.** Associations between the extent of transposable element (TE)-mediated H3K9me2 enrichment and genic H3K9me2 enrichment differ between genes close (distance to a TE is smaller than the 50% quantile, blue) and distant (gray) to TEs.

**Figure supplement 3.** Associations between the magnitude of transposable element (TE)-mediated H3K9me2 enrichment and gene expression rank (lower rank suggests higher expression) *do not* differ between genes close (distance to a TE is smaller than the 50% quantile, orange) and distant (gray) to TEs (see *Figure 3—figure supplement 4* for the extent of TE-mediated H3K9me2 enrichment).

**Figure supplement 4.** Associations between the extent of transposable element (TE)-mediated H3K9me2 enrichment and gene expression rank *do not* differ between genes close (distance to a TE is smaller than the 50% quantile, orange) and distant (gray) to TEs for most genomes.

**Figure supplement 5.** Associations between the magnitude (top) and extent (bottom) of transposable element (TE)-mediated H3K9me2 enrichment and genic H3K9me2 enrichment differ between genes close (distance to a TE is smaller than the 50% quantile, blue) and distant (gray) to TEs for both TEs 5′ and 3′ to genes.

**Figure supplement 6.** Associations between the magnitude (top) and extent (bottom) of transposable element (TE)-mediated H3K9me2 enrichment and gene expression rank (lower rank suggests higher expression) do *not* differ between genes close (distance to a TE is smaller than the 50% quantile, orange) and distant (gray) to TEs for TEs 5′ and 3′ to genes.

**Figure supplement 7.** z-Scores for comparing the H3K9me2 enrichment (left) and expression rank (right) of homologous genic alleles whose nearby transposable elements (TEs) with (blue/orange) or without (gray) epigenetic effects (as defined as the extent of H3K9me2 enrichment >1 kb; see *Figure 3C* for categorizing TEs with the magnitude of H3K9me2 enrichment, which gives similar results).

**Figure supplement 8.** z-Scores for comparing the H3K9me2 enrichment (left) and expression rank (right) of homologous genic alleles whose nearby 5′ or 3′ transposable elements (TEs) with (blue/orange) or without (gray) epigenetic effects as defined as the magnitude of H3K9me2 enrichment >1 (top) or the extent of H3K9me2 enrichment >1 kb (bottom).

**Figure supplement 9.** The log2 gene expression level (RPKM) for homologous alleles with (x-axis) and without (y-axis) nearby transposable elements (TEs).

**Figure supplement 10.** Z-scores for comparing the *magnitude* of transposable element (TE)-mediated H3K9me2 enrichment on the intergenic side and on the genic side for TEs close to (green) and far (gray) from genes whose expression is at least 10 RPKM.

**Figure supplement 11.** Z-scores for comparing the *extent* of transposable element (TE)-mediated H3K9me2 enrichment on the intergenic side and on the genic side for TEs close to (green) and far (gray) from genes whose expression is *smaller than* 10 RPKM.

**Figure supplement 12.** Z-scores for comparing the *extent* of transposable element (TE)-mediated H3K9me2 enrichment on the intergenic side and on the genic side for TEs 5′ (dark green) and 3′ (light green) to genes whose expression is *greater than* 10 RPKM.

**Figure supplement 13.** The extent of transposable element (TE)-induced H3K9me2 enrichment on the side facing insulator sequences CTCF (**A**) and BEAF-32 (**B**) or on the other side.

and 4 for the magnitude and extent of TE-mediated epigenetic effects, respectively) nor depend on gene-TE distance except in one genome (gene expression ~ TE epigenetic effects + distance to TE + interaction; p-value for interaction term >0.05 for all genomes except for *D. santomea,* p<0.05 for both magnitude and extent of the effects). These observations substantially differ from our observed prevalent associations between TE-mediated effects and genic epigenetic states across species (*Figure 3A*, *Figure 3—figure supplements 1 and 2*).

TEs upstream to genes (i.e., 5′ to genes) are expected to have greater potential to influence the promoters than those 3′ to genes. If TE-mediated epigenetic effects could indeed lower gene expression, the associations should be more likely observed with TEs 5′ to genes. Analyzing TEs 5′ and 3′ to genes together could thus potentially obscure the signal. To test this possibility, we first investigated whether TEs 5′ and 3′ to genes have differential impacts on the epigenetic states of genes. We still observed significant positive associations between the magnitude and extent of TE-mediated H3K9me2 enrichment and genic H3K9me2 enrichment for both TEs 5′ and 3′ to genes (*Spearman rank correlation coefficient* $\rho$ =0.13–0.38 [TEs 5′ to genes] and 0.16–0.43 [TEs 3′ to genes], p<0.05 for all tests except for the magnitude of the effect for TEs 5′ to genes, *D. yakuba* strain 2; *Figure 3B*). Also, there is no consistent trend on whether the correlation is stronger for TEs 5′ or 3′ to genes across genomes (*Figure 3B*). The dependency of this association on TE-gene distance also generally holds for both TEs 5′ and 3′ to genes (p-value for interaction term <0.05 for more than half of the genomes, *Figure 3—figure supplement 5*). On the other hand, we still observed nearly absent associations between TE-mediated epigenetic effects and gene expression rank even when analyzing TEs 5′ and 3′ to genes separately, and the few significant associations are all for TEs 5′ to genes (*Spearman rank correlation test*, p>0.05 for all tests except for TEs 5′ to genes, *D. yakuba* strain 1 [both magnitude and extent] and *D. yakuba* strain 2 [extent] *Figure 3B*). Again, the associations between TE-mediated epigenetic effects and gene expression do not depend on TE-gene distance either for TEs 5′ and 3′ to genes for most of the cases (regression analysis interaction term, p>0.05 for all genomes and for both TEs 5′ and 3′ to genes, except for the magnitude for TEs 5′ to genes [*D. simulans* strain 1] or for the extent for TE 3′ to genes [*D. mauritiana*], *Figure 3—figure supplement 6*).

The above analyses may have limited power because variation in gene expression levels within a genome could be due to intrinsic gene properties, instead of TE-mediated effects. To directly test the effects of TE-mediated H3K9me2 enrichment on nearby gene expression, we compared the expression of homologous alleles with and without adjacent TEs in *D. simulans* and *D. yakuba,* two species that we have epigenomic and transcriptomic data for two strains. We estimated z-scores, which compare H3K9me2 enrichment and expression rank between homologous alleles (see Materials and methods). A positive z-score indicates that the allele adjacent to TE insertions has higher H3K9me2 enrichment or expression rank (i.e., lower expression) than the homologous allele without a nearby TE. We again observed TEs with stronger epigenetic effects associated with higher z-scores of genic H3K9me2 enrichment, which confirms their impacts on genic epigenetic states (*Mann-Whitney U test,* p<0.05 for all comparisons; *Figure 3C* and *Figure 3—figure supplement 7* for categorizing genes according to the magnitude and extent of TE-induced H3K9me2 enrichment, respectively). By analyzing TEs 5′ and 3′ to genes, we observed a consistent trend that z-scores for genes near TEs with epigenetic effects are larger, even though the comparisons are only significant for a subset of comparisons, probably due to the reduced sample size (*Figure 3—figure supplement 8*). On the contrary, there is no association between TE-mediated epigenetic effects and z-scores of gene expression rank (*Mann-Whitney U test*, p>0.05 for all comparisons; *Figure 3C* and *Figure 3—figure supplement 7* for categorizing genes according to the magnitude and extent of TE-induced H3K9me2 enrichment, respectively; *Figure 3—figure supplement 8* for looking at TEs 5′ and 3′ to genes separately). Directly comparing the expression of homologous alleles with and without TEs reached similar conclusions (*Figure 3—figure supplement 9*). Overall, our results suggest that, across genomes, TE-mediated epigenetic effects lead to robust enrichment of heterochromatic marks at neighboring genes, which, contrary to expectation, does *not* have a predominantly negative impact on the expression of neighboring genes. It is worth noting that our analyses focus on identifying the *genome-wide average* pattern; it is still plausible that the local enrichment of H3K9me2 induced by individual TEs occasionally lowers nearby gene expression (e.g., genes with positive z-scores in *Figure 3C* or genes with TE-mediated regulation; *Ninova et al., 2020*). In fact, many early examples of TE-mediated epigenetic effects were discovered by the phenotypic consequences of reduced neighboring gene expression (reviewed in *Choi and Lee, 2020*).

The extent of local enrichment of repressive epigenetic marks for a handful of TE insertions was previously suggested to be influenced by the expression of neighboring genes in a mouse cell line (*Rebollo et al., 2012*). This observation suggests the possibility that the extent of TEs' epigenetic effects is, in return, restrained by the expression of neighboring genes. If true, this phenomenon may explain our observed lack of negative associations between TE-mediated epigenetic effects and

neighboring gene expression. To investigate this possibility on a genome-wide scale, we compared the H3K9me2 enrichment for a TE on the side that faces a gene with appreciable expression in 16–18 hr embryo (at least 10 RPKM; genic side) and the other side that does not face the gene (intergenic side; *Figure 3D*). We predict that the transcriptional effects of genes on TE-mediated enrichment of H3K9me2 should be the most prominent on the 'genic side' of a TE. Also, the differences between the two sides of a TE should be larger for TEs closer to genes.

To test these predictions, we estimated the normalized difference between TE-mediated H3K9me2 enrichment on the intergenic and genic sides by calculating a z-score. A positive z-score would mean that the magnitude or extent of TE-mediated H3K9me2 enrichment is more restricted on the genic side than on the intergenic side. Consistent with these predictions, there is a general trend that the z-score for the extent of spread is positive for TEs close to highly expressed genes (RPKM >10), although the comparisons are statistically significant only for a subset of the genomes (*Mann-Whitney U test,* p<0.001 for *D. simulans* strain 1 and *D. teissieri, Figure 3D*). On the other hand, z-scores for the extent of spread are not significantly different from 0 for TEs far from highly expressed genes (*Mann-Whitney U test,* p>0.05 for all genomes, *Figure 3D*) and are significantly larger for TEs close to highly expressed genes than for those far from highly expressed genes (*Mann-Whitney U test,* p<0.05 for *D. simulans* strain 1, *D. teissieri,* and *D. santomea; Figure 3D*). We also found significant negative associations between the z-score of a TE and its distance to the nearest gene (*Spearman rank correlation coefficient* $\rho$ =–0.17 [*D. mauritiana*], –0.13 [*D. simulans* strain 1], –0.11 [*D. teissieri*], and –0.10 [*D. santomea*], p<0.01 for all tests), further supporting that the restricted extent of TE-mediated H3K9me2 enrichment on the genic side depends on TE-gene distance. Curiously, comparisons based on the *magnitude* of TE-induced H3K9me2 enrichment did not find differences between the genic side and the intergenic side, nor between TEs that are close to or far from highly expressed genes (*Mann-Whitney U test* and *Spearman rank correlation test,* p>0.05 for all comparisons, *Figure 3— figure supplement 10*; also see Discussion).

If the restricted extent of TE-mediated H3K9me2 enrichment on the genic side indeed results from the transcription activities of neighboring genes, this effect should mainly be observed with TEs near highly, but not lowly, expressed genes and would be stronger for TEs 5' to genes (i.e., near promoters) than those 3' to genes. Consistent with these predictions, z-score for TEs close to lowly expressed genes (RPKM <10) are *not* significantly different from 0, nor do they differ between TEs close to or far from genes (*Mann-Whitney U test,* p>0.05 for both comparisons in all genomes, *Figure 3—figure supplement 11*). Interestingly, TEs 5' to highly expressed genes are more likely to have z-scores significantly larger than 0 (for four out of eight genomes) than those of TEs 3' to genes (for one out of eight genomes), suggesting a more restricted extent of H3K9me2 enrichment on the genic side for TEs near promoters (*Mann-Whitney U test,* p<0.05 for *D. simulans* strain 1, *D. teissieri, D. santomea,* and *D. yakuba* strain 2 [TEs 5' to genes] and for *D. teissieri* [TEs 3' to genes], *Figure 3—figure supplement 12*). Similarly, the negative associations between the z-score of a TE and its distance to the nearest gene are significant for TEs 5' to genes in four genomes (*Spearman rank correlation coefficient* $\rho$ =–0.24 [*D. mauritiana*], –0.18 [*D. simulans* strain 1], –0.11 [*D. teissieri*], and –0.17 [*D. santomea*], p<0.05 for all tests) but not for those 3' to genes (*Spearman rank correlation tests,* p>0.05 for all tests). Such observation is consistent with the idea that the distance-dependent effect of gene transcription on the extent of TE-mediated epigenetic effects only holds for TEs near promoters. Overall, our findings reveal a complex relationship between TE-mediated epigenetic effects and neighboring gene expression, and strongly suggest that the extent of TE-mediated H3K9me2 enrichment is influenced by the expression of adjacent genes, especially on the side of a TE that faces the promoter of a nearby, highly expressed gene.

## TEs with epigenetic effects are selected against across species

TEs exerting epigenetic effects were previously suggested to experience stronger purifying selection than other TEs (reviewed in *Choi and Lee, 2020*). If TE-mediated epigenetic effects indeed impair host fitness and are thus selected against, such effects could play important roles in shaping the evolution of both TEs and their host genomes. Yet, previous analyses testing the presence of selection against TEs with epigenetic effects were restricted to few model species. More importantly, the confounding effects of other deleterious mechanisms of TEs (e.g., ectopic recombination; see below)

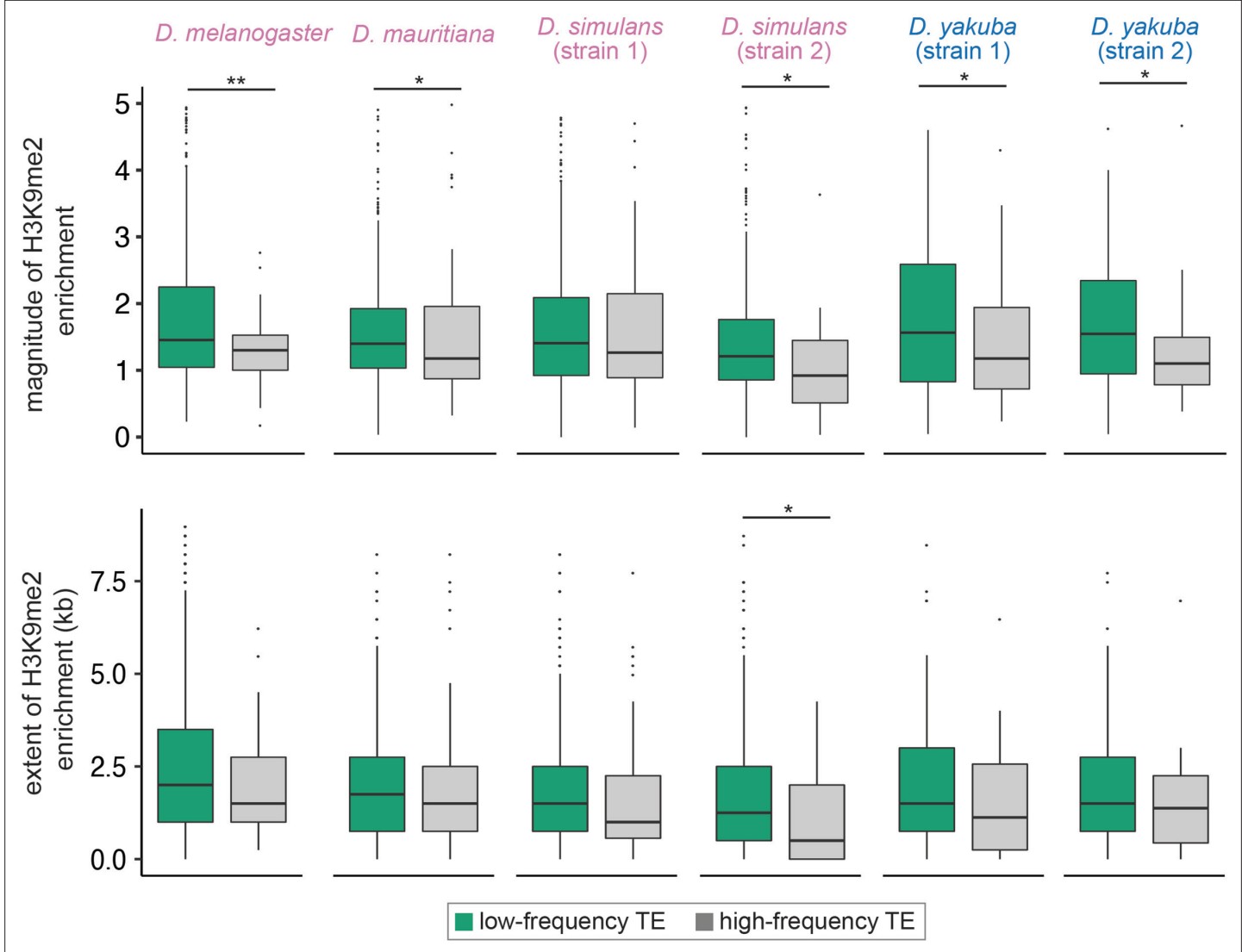

**Figure 4.** Associations between the population frequencies and epigenetic effects of transposable elements (TEs). The magnitude (top) and extent (bottom) of TE-mediated enrichment of H3K9me2 for low-frequency TEs (usually considered as strongly selected, green) and high-frequency TEs (gray) in *Drosophila melanogaster, Drosophila mauritiana, Drosophila simulans,* and *Drosophila yakuba.* Low-frequency TEs are those that are only found in the focused genome, while high-frequency TEs are identified in both the focused genome and the population of genomes. *Mann-Whitney U test,* \*\*p<0.01, \*p<0.05.

on TE population frequencies could not be ruled out in those studies (*Hollister and Gaut, 2009*; *Lee, 2015*; *Lee and Karpen, 2017*).

To investigate the fitness impacts of TE-mediated H3K9me2 enrichment, we estimated the frequencies of TEs that are included in our analysis in four species, using previously published populations of genomes (*D. melanogaster* [*Lack et al., 2015*], *D. simulans* and *D. yakuba* [*Rogers et al., 2014*], and *D. mauritiana* [*Garrigan et al., 2014*]). Population frequencies of TEs are strongly influenced by the strength of natural selection against their deleterious effects (reviewed in *Charlesworth and Langley, 1989*; *Lee and Langley, 2010*; *Barrón et al., 2014*), with low-frequency TEs generally expected to be more deleterious than high-frequency TEs. We categorized TEs in our focused genome into high-frequency and low-frequency according to whether they are (high-frequency) or are not (i.e., singletons; low-frequency) identified in the population samples. We found that the *magnitude* of TE-induced H3K9me2 enrichment is significantly higher for low-frequency TEs than for high-frequency TEs in all four species (*Mann-Whitney U test,* p<0.05 for all but *D. simulans* strain 1, *Figure 4*), which extends previous investigations in *D. melanogaster*. Intriguingly, the difference is much weaker when

**Table 1.** Logistic regression coefficients for the effects of transposable element (TE) length and TE-mediated H3K9me2 enrichment (magnitude and extent) on the population frequencies of TEs.

| | Magnitude of H3K9me2 enrichment | | Extent of H3K9me2 enrichment | |
|---|---|---|---|---|
| | TE length | Magnitude | TE length | Extent |
| *D. melanogaster* | –4.95E-04 | **–4.59E-01\*** | –5.22E-04 | **–8.63E-05** |
| *D. mauritiana* | –1.83E-03 | **–2.74E-02** | –1.87E-03 | 1.55E-05 |
| *D. simulans* (strain 1) | –2.56E-04 | 1.07E-01 | –2.52E-04 | 9.00E-06 |
| *D. simulans* (strain 2) | –3.13E-03 | **–2.50E-01** | –3.21E-03 | **–2.41E-04** |
| *D. yakuba* (strain 1) | –3.63E-04 | **–1.68E-01** | –3.73E-04 | **–7.77E-05** |
| *D. yakuba* (strain 2) | –4.66E-04 | **–1.27E-01** | –4.72E-04 | **–2.87E-04** |

Negative regression coefficients for TE-mediated epigenetic effects on TE population frequencies are in bold. \*p<0.05.

comparing the *extent* of TE-mediated H3K9me2 enrichment, and the comparison is significant for only one genome (*Mann-Whitney U test,* p<0.05 for strain 2 of *D. simulans,* but >0.05 for all other genomes; *Figure 4,* see Discussion).

A potential confounding factor for our observed associations between TEs' epigenetic effects and population frequencies is TE length. TE length was previously observed to negatively correlate with population frequencies of TEs (*Petrov et al., 2003*; *Petrov et al., 2011*), either because of the larger potential of long TEs in disrupting functional elements or their higher propensity to be involved in deleterious ectopic recombination (*Petrov et al., 2003*). Because the strength of TE-mediated epigenetic effects also positively correlated with TE length (*Figure 1—figure supplement 3*), our observed negative associations between TE frequencies and epigenetic effects could instead result from other harmful effects of TEs. To investigate this possibility, we performed logistic regression analysis to test the effects of TE-mediated H3K9me2 enrichment on population frequencies while accounting for the influence of TE length (population frequency ~ length + TE epigenetic effects). Because of the co-linearity between predictor variables in the regression model (TE length and epigenetic effects, *Figure 1—figure supplement 3*), this analysis is expected to have restricted statistical power. Nevertheless, regression coefficients for the magnitude of H3K9me2 enrichment are negative for all but one genome (*Table 1*), and the coefficient is significantly negative for *D. melanogaster*.

Local meiotic recombination rate is another factor that could potentially confound our observed negative associations between TE population frequencies and epigenetic effects. If TEs with weaker epigenetic effects tend to locate in genomic regions with low recombination rate, lower probability of ectopic recombination (*Langley et al., 1988*), or reduced efficacy of selection (*Hill and Robertson, 1966*; *Felsenstein, 1974*; reviewed in *Charlesworth and Langley, 1989*; *Lee and Langley, 2010*; *Barrón et al., 2014*; *Kent et al., 2017*) may instead drive their higher population frequencies. However, we found no associations between TE-mediated epigenetic effects and local recombination rate in *D. melanogaster,* the species with a high-resolution recombination map (*Comeron et al., 2012*) (*Spearman rank correlation tests*, p=0.59 [magnitude] and 0.98 [extent]). Such observation is consistent with a previous analysis using different *D. melanogaster* strains (*Lee and Karpen, 2017*) and suggests that the difference in meiotic recombination rate unlikely confounded our analysis of TE population frequencies.

The average deleterious effects of TE insertions were found to differ between TE families (reviewed in *Charlesworth and Langley, 1989*; *Barrón et al., 2014*), which could also confound our analysis, given that the strength of TE-mediated epigenetic effects varies along the same axis (*Figure 2*). Accordingly, we studied the associations between TE-induced H3K9me2 enrichment and TE population frequencies among *copies of the same TE family within species*, aiming to exclude the potential confounding effects of family identity on TE population frequencies. We performed logistic regression analysis for individual TE families that have at least 10 insertions while accounting for the effects of TE length (population frequency ~ TE epigenetic effects + length). With collinearity among predictor variables (see above) and the small number of TEs included for each TE family (fewer than 30 copies

for 19 out of 23 TE families, with a median of 19 of TEs included in the regression analysis), these analyses are again underpowered. For most of the TE families tested, we observed negative regression coefficients for the magnitude of TE-mediated H3K9me2 enrichment (*Table 2*), which is more than expected by chance (18 out of 23 TE families tested, *binomial test,* p=0.0106). We observed similar results with the extent of TE-mediated H3K9me2 enrichment (17 out of 23 TE families tested, *binomial test,* p=0.0346, *Table 2*). Despite the lack of statistical power, H and Cr1a families showed significant negative regression coefficients for TE-mediated epigenetic effects in one of the *D. yakuba* strains. Overall, our results revealed that, after controlling for the effects of TE length and family identity, we still found negative associations between the strength of TE-mediated epigenetic effects and TE population frequencies. By excluding the potential impacts of confounding factors on TE population frequencies, our observation strongly supports that TE-mediated enrichment of repressive marks is disfavored by natural selection in multiple species.

## Epigenetic effects of TEs negatively associate with genomic TE abundance across species

The magnitude and extent of TE-mediated H3K9me2 enrichment vary not only within genomes and between individuals, but also between species (*Figure 1*). According to a previously proposed hypothesis (*Lee and Karpen, 2017*), this between-species difference in the strength of TE-mediated H3K9me2 enrichment could determine the average strength of selection removing TEs, leading to species-specific TE abundance in the euchromatic genome. To test this prediction, we investigated the associations between euchromatic TE numbers and the average magnitude and extent of TE-induced local enrichment of H3K9me2. Because the assignment of TEs into families is biased against TEs in *yakuba* complex species (*Figure 2—figure supplement 3*), all TEs, irrespective of whether we could assign their family identity, were included in this between-species analysis (see Materials and methods).

We found that TEs in species of the *yakuba* complex show a much larger magnitude of H3K9me2 enrichment than those in species of the *melanogaster* complex (*Figure 5A*, *Mann-Whitney U test,* p=0.029). After controlling for the strong impacts of species complex, we found significant negative associations between the number of euchromatic TEs and the average magnitude of TE-mediated H3K9me2 enrichment across genomes (*Figure 5A*, TE abundance ~ species complex + epigenetic effects; regression coefficient for the magnitude of H3K9me2 enrichment: –1603.5; ANOVA F-value: 9.05, df = 1, p=0.030). Exclusion of *D. simulans* strain 2, a sample that shows lower H3K9me2 enrichment consistency between replicates (see Materials and methods) gave consistent results (regression coefficient: –1602; ANOVA F-value: 5.26, df = 1, p=0.080). On the other hand, the extent of TE-mediated H3K9me2 spreading does not differ between species complexes (*Mann-Whitney U test,* p=0.89) nor associate with the number of euchromatic TEs (*Figure 5A*, regression coefficient for the extent of H3K9me2 enrichment [kb]: 925; ANOVA F-value: 0.03, df = 1, p=0.87; excluding *D. simulans* strain 2 – regression coefficient: 1028, F-value: 0.13, df = 1, p=0.78; see Discussion). This finding echoes our observations of the nearly absent associations between the extent of TE-mediated H3K9me2 enrichment and TE population frequencies (*Figure 4*). It is worth noting that the associations between the magnitude, but not the extent, of TE-mediated H3K9me2 enrichment and genomic TE abundance was also documented in a much smaller study previously (*Lee and Karpen, 2017*). To account for the phylogenetic non-independence among species, we used phylogenetic generalized least squares (PGLS, *Grafen, 1989*; *Martins and Hansen, 1997*) to repeat the regression analysis and found consistent results (regression coefficient for the magnitude of H3K9me2 enrichment: –1948; ANOVA F: 7.49, df = 1, p=0.034; regression coefficient for the extent of H3K9me2 enrichment [kb]: 599; ANOVA F: 0.14, df = 1, p=0.72). Overall, the negative associations between the magnitude of TE-mediated H3K9me2 enrichment and abundance of euchromatic TEs within species complex support our prediction that varying strength of TE-mediated epigenetic effects could drive between-species differences in genomic TE abundance on a short evolutionary time scale.

## Species-specific repressive chromatin landscape associates with between-species differences in epigenetic effects of TEs

If varying strength of TE-mediated epigenetic effects indeed contributes to between-species differences in euchromatic TE copy number, as suggested by our observations (*Figure 5A*), species-specific

**Table 2.** Logistic regression coefficients for the effects of transposable element (TE) length and TE-mediated H3K9me2 enrichment (magnitude and extent) on the population frequencies of TEs from different families.

| TE family | Magnitude of H3K9me2 enrichment | | Extent of H3K9me2 enrichment | |
|---|---|---|---|---|
| | TE length | Magnitude | TE length | Extent |
| *D. melanogaster* | | | | |
| BS | –8.10E-03 | **–2.65E+00** | –5.18E-03 | **–5.16E+02** |
| 297 | –4.86E-04 | **–1.43E+00** | –5.12E-04 | **–1.10E-04** |
| jockey | 2.17E-04 | **–1.30E+00** | 2.50E-04 | **–1.42E-03** |
| pogo | 9.46E-04 | **–1.04E+00** | 9.18E-04 | 9.47E-05 |
| Doc | 2.65E-04 | 2.20E-01 | 2.96E-04 | 2.55E-04 |
| hopper | –1.53E-04 | 1.36E+00 | –1.63E-04 | **–2.14E-04** |
| | | | | |
| *D. mauritiana* | | | | |
| HB | –2.94E-03 | **–2.10E-01** | –2.96E-03 | **–3.23E-05** |
| Bari | –4.78E-03 | 6.58E-01 | –4.96E-03 | 3.65E-04 |
| hopper | 6.66E-03 | **–1.96E+00** | 8.70E-03 | **–3.72E-04** |
| | | | | |
| *D. simulans (strain 1)* | | | | |
| H | 1.84E-02 | **–9.70E-02** | 1.76E-02 | **–3.08E-04** |
| transib2 | –3.69E-03 | **–1.26E+00** | –3.71E-03 | **–3.08E-04** |
| Tc1 | –1.19E-01 | **–3.56E-01** | –1.16E-01 | **–2.33E-04** |
| 1,360 | –3.84E-02 | 1.37E-01 | –4.02E-02 | 8.88E-05 |
| diver2 | 2.16E-04 | **–1.39E-01** | –2.54E-04 | **–1.39E-03** |
| Helena | –3.99E-03 | 2.85E-01 | –3.62E-03 | 4.20E-04 |
| HB | –2.45E+00 | **–4.26E+00** | –2.48E+00 | **–1.61E-03** |
| roo | –4.01E-04 | **–2.71E-01** | –5.67E-04 | 4.10E-04 |
| | | | | |
| *D. simulans (strain 2)* | | | | |
| H | –4.97E-03 | **–6.63E-01** | –4.94E-03 | **–3.02E-05** |
| Tc1 | –2.28E-03 | **–8.68E-01** | –2.46E-03 | **–7.59E-04** |
| | | | | |
| *D. yakuba (strain 1)* | | | | |
| H | –1.80E-03 | **–3.79E-01** | –1.50E-03 | **–2.75E-04** |
| Cr1a | 9.53E-05 | **–6.66E-01** | –1.05E-04 | **–3.35E-04** |
| | | | | |
| *D. yakuba (strain 2)* | | | | |
| H | –2.29E-02 | **–3.89E-01** | –5.48E-02 | **–8.98E-03*** |
| Cr1a | 1.32E-03 | **–1.32E+00*** | 4.42E-04 | **–1.27E-04** |

Negative regression coefficients for TE-mediated epigenetic effects on TE population frequencies are in bold. *$p < 0.05$.

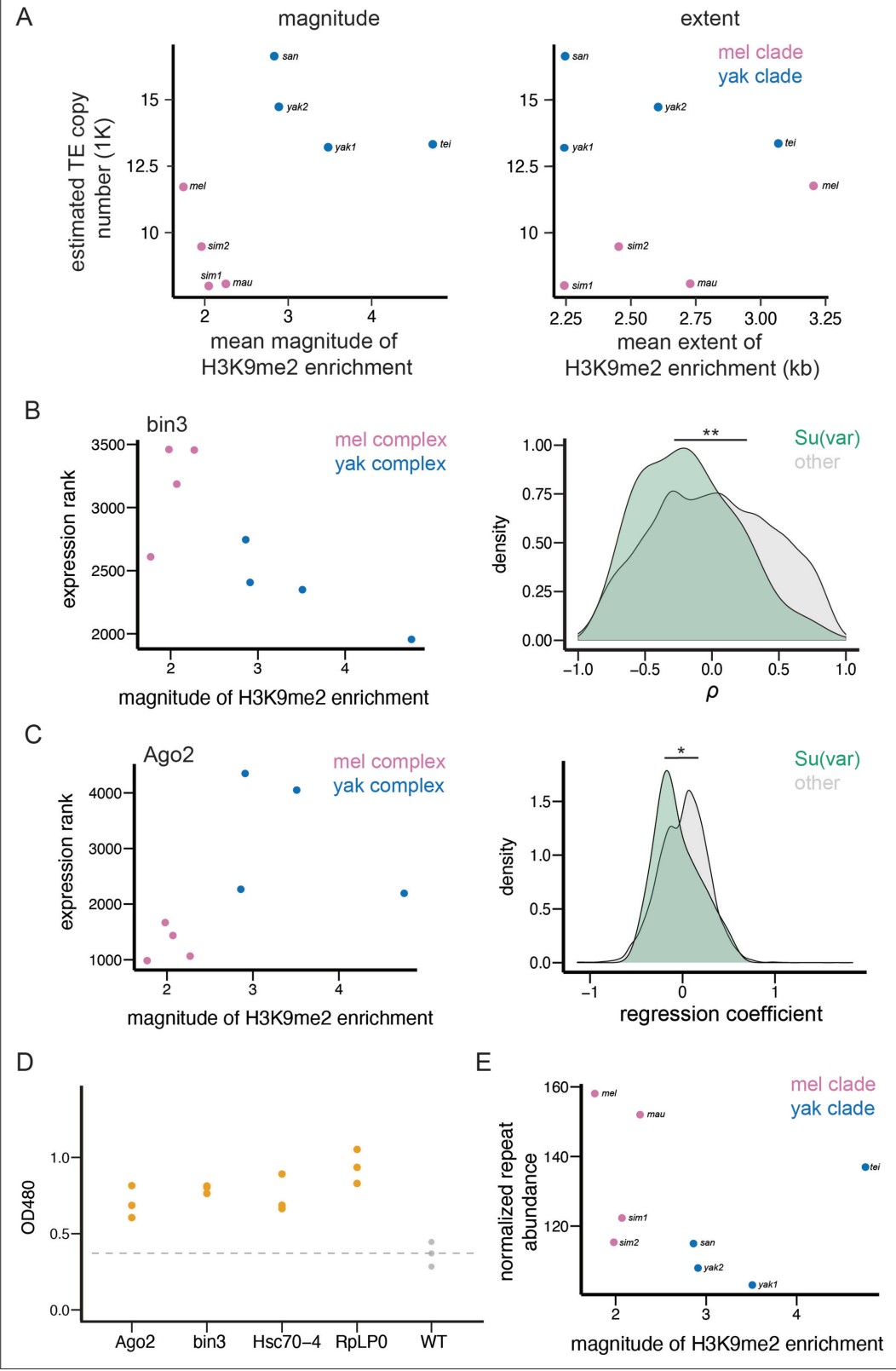

**Figure 5.** Transposable element (TE)-mediated epigenetic effects associate with genomic TE abundance and repressive chromatin landscape. (**A**) The associations between the mean magnitude (left)/extent (right) of TE-mediated enrichment of H3K9me2 and the number of estimated euchromatic TEs across species. (**B**) An example gene (*bin3*) whose expression rank negatively correlates with the mean magnitude of TE-mediated H3K9me2

*Figure 5 continued on next page*

*Figure 5 continued*

enrichment across species (left). The distributions of *Spearman rank correlation coefficient* ( $\rho$ ) between the magnitude of TE-mediated H3K9me2 enrichment and the expression ranks significantly differ between *Su(var)* genes (green) and other genes in the genome (gray; right). (**C**) An example gene (*Ago2*) whose expression rank negatively correlates with the magnitude of TE-mediated H3K9me2 enrichment between species *within species complex,* but differs significantly between species complex (left). The distributions of *regression coefficients* for the effect of a gene's expression on the magnitude of TE-mediated H3K9me2 enrichment significantly differs between *Su(var)* genes (green) and other genes in the genome (gray, right). (**D**) The reduced dosage of *Su(var)* genes influences the epigenetic silencing effect of 1360. For each candidate *Su(var)*, we performed three replicates (three independent crosses), and one dot represents one cross. (**E**) The mean magnitude of TE-mediated enrichment of H3K9me2 associates with the abundance of H3K9me2-enriched Kmers. \*\*p<0.01 and \*p<0.05 for *Mann-Whitney U test* or *Kolmogorov-Smirnov test* (see text).

differences in genetic factors that modulate the epigenetic effects of TEs could be the ultimate drivers for varying genomic TE abundance. Consistent with this possibility, we observed that the strength of TE-mediated epigenetic effects strongly depends on the host genetic background (***Figure 2***). The similarities between TE-mediated enrichment of repressive marks and the well-studied 'position effect variegation'' (PEV, ***Gowen and Gay, 1934***) suggest a feasible path to narrow down the causes for variation in the epigenetic effects of TEs. Repressive epigenetic marks at constitutive heterochromatin are long known to spread to juxtaposed euchromatic genes, a phenomenon known as PEV (reviewed in ***Elgin and Reuter, 2013***). Several genetic factors have been identified to modulate the strength and extent of PEV (reviewed in ***Girton and Johansen, 2008***; ***Elgin and Reuter, 2013***). These include *Su(var)* genes, whose protein products act as structural or enzymatic components of the heterochromatin (***Girton and Johansen, 2008***; ***Elgin and Reuter, 2013***; ***Swenson et al., 2016***), and heterochromatic repeats, which are targets of heterochromatin formation (***Girton and Johansen, 2008***; ***Elgin and Reuter, 2013***). The ratio between the dosage of these two players (targets and regulators) was postulated to influence the nucleation and formation of constitutive heterochromatin and, accordingly, the spread of repressive marks from constitutive heterochromatin to the euchromatic genome (***Locke et al., 1988***).

In order to test the hypothesized role of *Su(var)s* and heterochromatic repeats in shaping between-species differences in TE-mediated epigenetic effects, multi-species comparisons that span a reasonable phylogenetic resolution would be needed. Also, it would be important to demonstrate that *Su(var)s* that regulate the spreading of heterochromatic marks from constitutive heterochromatin play a similar role in the epigenetic effects of TEs in the euchromatic genome. This is because, though both enriched with H3K9me2/3, heterochromatin is comprised of large blocks of repetitive sequences enriched for repressive epigenetic marks (***Riddle et al., 2011***), whereas euchromatic TEs, though epigenetically silenced, are relatively short (***Lee, 2015***; ***Lee and Karpen, 2017***) and usually surrounded by sequences enriched for active epigenetic marks (***Kharchenko et al., 2011***).

To test our hypothesis that species-specific differences in genetic factors contribute to varying epigenetic effects of TEs, we investigated whether there are positive associations between TE-mediated epigenetic effects and the dosage of *Su(var)* genes, which promotes the spreading of repressive marks from constitutive heterochromatin (***Girton and Johansen, 2008***; ***Elgin and Reuter, 2013***; ***Swenson et al., 2016***). We used the expression of *Su(var)* genes in 16–18 hr embryos, a developmental stage matching our epigenomic data, as a proxy for the dosage of Su(var) protein products. Because we only found that the magnitude, but not the extent, of TE-mediated H3K9me2 enrichment negatively associates with TE copy number (***Figure 5A***), our following analysis focused on the magnitude of TEs' epigenetic effects, with the aim of identifying the ultimate cause for varying genomic TE abundance. We estimated the *Spearman rank* correlation coefficients ( $\rho$ ) between the expression rank of a gene (rank from the highest expressed genes) and the average magnitude of TE-mediated H3K9me2 enrichment across genomes (see ***Figure 5B*** for an example, also see Materials and methods). A negative correlation indicates that a *Su(var)* gene has higher expression (and thus lower expression rank) in a genome with stronger epigenetic effects of TEs. We found that *Su(var)* genes, as a group, have significantly lower $\rho$ than other genes in the genome (*Mann-Whitney U test*, p=0.0073) and have a shifted distribution of $\rho$ toward smaller values (*Kolmogorov-Smirnov test*, p=0.033, ***Figure 5B***). These observations suggest that the expression levels of *Su(var)s* correlate more positively with the magnitude of TE-mediated H3K9me2 enrichment than other genes in the genome. Among analyzed

*Su(var)s*, the $\rho$ for *bin3*, promoter of small-RNA-mediated silencing (**Singh et al., 2011**), is among the top 5% of all genes. *Lhr* and *HP4*, both of whose protein products are structural components of heterochromatin (**Greil et al., 2007**), and *Hsc70-4*, whose protein product is an interactor of core heterochromatin protein HP1a (**Swenson et al., 2016**), are among top 10% genome-wide (**Supplementary file 1**).

Intriguingly, the expression rank of *Ago2*, a key gene in initiating epigenetic silencing and a *Su(var)* (**Deshpande et al., 2005**), is among the top 10% genome-wide that *positively* correlates with TE-mediated enrichment of H3K9me2 (**Supplementary file 1**), an association that is opposite to prediction. Upon further examination, we found that the expression of *Ago2* significantly differs between species in the two species complexes, which drives the positive correlation (**Figure 5C**). Accordingly, we performed regression analyses that associate the magnitude of TE-mediated H3K9me2 enrichment and gene expression rank while accounting for the effects of species complex (magnitude of TE-mediated epigenetic effects ~ gene expression + species complex). Similar to the analysis based on $\rho$ (see above, **Figure 5B**), the regression coefficients are significantly smaller for *Su(var)s* than those for other genes (*Mann-Whitney U test,* p=0.063; *Kolmogorov-Smirnov test,* p=0.032; **Figure 5C**), further corroborating our findings. With the regression analysis, we found that the association between *Ago2* expression rank and the magnitude of TE-mediated epigenetic effects is negative and is among 10% genome-wide. We also found other *Su(var)* genes whose regression coefficients are among the top 10% of all genes, which includes *RpLP0*, whose protein product is a core interactor of HP1a (**Frolov and Birchler, 1998**), and *Su(var)3–3*, which codes for an eraser of active histone modification (**Rudolph et al., 2007**, **Supplementary file 1**).

To confirm that these identified candidate *Su(var)s* also modulate TE-mediated local enrichment of H3K9me2 in the euchromatic genomes, we leveraged a previously published reporter system that allows the quantification of TE-mediated epigenetic effects (**Sentmanat and Elgin, 2012**). In this system, a DNA-based TE, *1360*, results in the enrichment of H3K9me2/3 at the immediate adjacent *mini-white* reporter gene in the euchromatic genome. The same study also found associations between the presence of *1360,* the enrichment level of H3K9me2 at the *mini-white* reporter gene, and the reduced amounts of red-eye pigmentation (**Sentmanat and Elgin, 2012**). Accordingly, the eye pigmentation level in this system serves as a convenient readout for the *magnitude* of TE-mediated H3K9me2 enrichment. By using existing loss-of-function *Su(var)* mutants, we found that reduced dosage (hemizygous) of candidate *Su(var)s* leads to significantly elevated levels of eye pigmentation, or reduced TE-mediated silencing effects (**Figure 5D**), supporting the roles of candidate *Su(var)s* in modulating the epigenetic effects of euchromatic TEs. With the assumption that the functional roles of *Su(var)* genes are conserved across *Drosophila* species studied, our findings suggest that the observed species-specific expression of *Su(var)s* could drive between-species differences in TE-mediated H3K9me2 enrichment in the euchromatic genome.

We also investigated whether the epigenetic effects of TEs negatively associated with the abundance of heterochromatic repeats, which was found to weaken the spreading of H3K9me2/3 from constitutive heterochromatin (reviewed in **Girton and Johansen, 2008**; **Elgin and Reuter, 2013**). We identified Kmers that are enriched with H3K9me2 in our ChIP-seq data and quantified their abundance using Illumina sequencing with PCR-free library preparation, in an effort to avoid biases in quantifying simple repeats (**Wei et al., 2018**) (see Materials and methods). Consistent with the prediction that repressive chromatin landscape weakens with increased abundance of heterochromatic repeats, we found that the magnitude of TE-mediated H3K9me2 enrichment is negatively associated with the abundance of H3K9me2-enriched repeats between species, though the comparison is not significant (**Figure 5E**, *Spearman rank correlation coefficient* $\rho$ =–0.595, p=0.13). Overall, our observations support the prediction that species-specific repressive chromatin landscape, which is shaped by the expression level of *Su(var)* genes and abundance of heterochromatic repeats, associates with differences in TE-mediated epigenetic effects between species.

## Discussion

The replicative nature of TEs has made them successful at occupying nearly all eukaryotic genomes. Yet, their 'success', or genomic abundance, drastically varies across the phylogenetic tree (**Huang et al., 2012**; **Elliott and Gregory, 2015**; **Wells and Feschotte, 2020**) and between closely related species (**Hu et al., 2011**; **Rius et al., 2016**; **Legrand et al., 2019**), raising important questions regarding the

evolutionary causes and functional consequences of varying genomic TE abundance. It is worth noting that most studies about the evolution of TE abundance, including ours, mainly concern insertions in the euchromatic genome. Although TEs are highly abundant in the heterochromatic genome, these TEs are typically fragmented (*Hoskins et al., 2007*; *Hoskins et al., 2015*) and would no longer be part of the 'life cycle' of TEs. The role of euchromatic TEs in determining the abundance and evolutionary dynamics of TEs in the heterochromatic genome is still a largely unaddressed question, mainly due to the challenges associated with assembling TEs in repeat-rich heterochromatic sequences (*Salzberg and Yorke, 2005*; *Treangen and Salzberg, 2011*, but see *Khost et al., 2017*; *Chang and Larracuente, 2019*; *Chang et al., 2019*).

Several hypotheses have been proposed to explain the large variation in euchromatic TE abundance. Phylogenetic signals may explain some of the differences between distantly related taxa (*Wells and Feschotte, 2020*) and, sometimes, between species within a taxa (*Szitenberg et al., 2016*). Variation in TE activities was also postulated to contribute to the wide variability of TE abundance (*Chen et al., 2019*; *Wong et al., 2019*, but see *Ho et al., 2021*). Still another plausible cause is systematic differences in the strength of natural selection removing TEs between species. Investigations of this hypothesis have been largely focused on population genetic parameters that influence the efficacy of natural selection, such as mating systems (*Wright and Schoen, 1999*; *Dolgin et al., 2008*; *Boutin et al., 2012*; *Arunkumar et al., 2015*; *Agren et al., 2014*) and effective population size (*Lynch and Conery, 2003*; *Mérel et al., 2021*). Here, our multi-species study tested step-by-step yet another evolutionary mechanism by which the strength of selection removing TEs could differ—through differences in the epigenetic effects of TEs that are driven by species-specific host chromatin landscape.

In this study, we investigated the prevalence, variability, and evolutionary importance of 'the epigenetic effects of TEs'—TE-mediated local enrichment of repressive epigenetic marks—in the euchromatic genomes of six *Drosophila* species. These species come from two important and well-studied complexes within the *melanogaster* species subgroup (*melanogaster* and *yakuba* complexes), providing good phylogenetic resolution to decipher the evolutionary role of epigenetic mechanisms in shaping species-specific TE abundance. We observed that TE-mediated local enrichment of repressive marks is prevalent in all species studied and widely varies both within and between genomes. Interestingly, in addition to the intrinsic biological properties of TEs, our analyses revealed that host genetic background plays a critical role in determining the strength of TE-mediated epigenetic effects. These TE-mediated enrichments of repressive marks alter the epigenetic states of neighboring euchromatic sequences, including actively transcribing genes. Importantly, TEs exerting such epigenetic effects are selected against across multiple species, and the strength of this TE-mediated epigenetic effect negatively correlates with genomic TE abundance within species complex. Our findings extend previous studies based on few species and provide one of the first support for the importance of the inadvertent harmful effects of TE epigenetic silencing in shaping divergent genomic TE landscapes.

Curiously, we only found a negative association between genomic TE abundance and the *magnitude* of TE-mediated H3K9me2 enrichment across species, but not the *extent* of the spreading, even though these two indexes of a TE significantly correlate within genomes (*Figure 1—figure supplement 2*). Similarly, the evidence for selection against TE-mediated epigenetic effects is stronger for the magnitude of TE-induced H3K9me2 enrichment than for the extent (*Figure 4*). Previous empirical (*Belyaeva and Zhimulev, 1991*; *Talbert and Henikoff, 2000*) and theoretical (*Erdel and Greene, 2016*) studies have suggested that the spreading of repressive heterochromatic marks is a nonlinear process. Accordingly, the extent of the spreading of repressive marks from TEs could be sensitive to the genomic context and subject to substantial stochasticity. This echoes our observations that the transcription of genes influences the extent of TE-mediated H3K9me2 spreading, but not the magnitude of H3K9me2 enrichment (*Figure 3D* and *Figure 3—figure supplement 10*; also see below). The magnitude of TE-mediated H3K9me2 enrichment, which is estimated in windows right next to TE boundary, may thus be a more direct measurement of the strength of TE-mediated epigenetic effects and more indicative of the associated harmful impacts.

One of our most surprising findings is perhaps the limited evidence in supporting that TE-mediated epigenetic effects reduce neighboring gene expression on a genome-wide scale. While TE-mediated epigenetic effects increase the enrichment of repressive epigenetic marks at adjacent genes, we found limited associations between such effects and gene expression within genomes (*Figure 3A and B*). Comparing the expression of homologous alleles with and without TEs showing epigenetic

effects also reached the same conclusion (*Figure 3C*). These observations echo previously reported lack of genome-wide associations between TE-induced enrichment of repressive epigenetic marks and the expression of neighboring genes in several model organisms (*Quadrana et al., 2016*; *Stuart et al., 2016*; *Lee and Karpen, 2017*; *Choi and Purugganan, 2018*) and are consistent with the wider observation that TEs upstream or downstream to genes are *not* predominantly associated with reduced gene expression (*Goubert et al., 2020*; *Ullastres et al., 2021*; *Rech et al., 2022*, reviewed in *Kelleher et al., 2020*; *Choi and Lee, 2020*). The limited impacts of TE-mediated H3K9me2 enrichment on gene expression could have resulted from the complex relationship between repressive epigenetic modification and gene expression (*de Wit et al., 2007*; *Yasuhara and Wakimoto, 2008*; *Riddle et al., 2011*; *Meng et al., 2016*; *Caizzi et al., 2016*), the varying sensitivity of genes to the enrichment of repressive marks (*Rudolph et al., 2007*; *Vogel et al., 2009*; *Riddle et al., 2011*), and the presence of other types of variants that also modulate gene expression (*Stranger et al., 2007*). Interestingly, our findings suggested another possibility—the transcription of genes, in return, influences TE-mediated epigenetic effects across genomes. Specifically, we found a more restricted spreading of repressive marks from TEs on the side facing a gene than on the intergenic side, and this difference is mainly observed for TEs near the promoters of highly expressed genes (*Figure 3—figure supplement 12*). It is worth noting that this difference in the extent of H3K9me2 spreading between the two sides of a TE is unlikely driven by the presence of insulator sequences (*Gaszner and Felsenfeld, 2006*) because of the limited associations between the presence of insulator sequences and the reduction in TE-mediated H3K9me2 spreading on the genic side (*Figure 3—figure supplement 13*). On the other hand, active histone modifications enriched at transcriptionally active genes could antagonize the assembly of heterochromatin (*Allshire and Madhani, 2018*), potentially restraining the spreading of repressive marks from silenced TEs. Consistently, in mice, active histone modification was reported to spread from an actively transcribing candidate gene into an adjacent TE (*Rebollo et al., 2012*). Similarly, active gene transcription at *Drosophila miranda* neo-Y chromosome was found to impede the formation of heterochromatin (*Wei et al., 2020*). Future analysis of the distributions of active histone modifications may help reveal the mechanistic cause for our observed genome-wide dependencies of the extent of TE-mediated epigenetic effects on the transcriptional activities of adjacent euchromatic genes.

If not due to reducing the expression of adjacent genes, what functional consequences of TE-mediated epigenetic effects could have impaired host fitness and led to the observed selection against these TEs? Euchromatic TEs enriched with repressive epigenetic marks were reported to spatially interact with constitutive heterochromatin through phase separation mechanisms, a process observed to alter 3D structures of genomes and inferred to lower host fitness (*Lee et al., 2020*). In addition, the epigenetic effects of TEs could shift the usual DNA repair process in the gene-rich euchromatic genome, perturbing the maintenance of local genome integrity. This is because double-stranded breaks happening in constitutive heterochromatin are repaired through a distinct cellular process from those in the euchromatic genome, owing to the enrichment of repressive epigenetic modifications (*Chiolo et al., 2011*; *Janssen et al., 2016*; *Janssen et al., 2019*). Still, the variance of gene expression was shown to be shaped by natural selection (*Metzger et al., 2015*; *Duveau et al., 2018*). Given the variegating properties of the spreading of repressive marks (reviewed in *Elgin and Reuter, 2013*), TE-mediated epigenetic effects could have shifted the variance, instead of the mean, of neighboring gene expression, impacting host fitness.

Building upon our findings of the prevalence and variability of TE-mediated epigenetic effects and selection against such effects, our multi-species analysis provides strong support for the previously proposed role of TE-mediated epigenetic effects in determining genomic TE abundance (*Figure 6*). This interpretation is based on the assumption that TE copy number is at equilibrium and transposition rates of TEs are similar across species, leaving the strength of selection removing TEs being the major determinant of genomic TE abundance (*Charlesworth and Charlesworth, 1983*). By combining transcriptomic analysis and *Drosophila* genetics experiment, we further revealed that between-species differences in TE-mediated epigenetic effects could be driven by species-specific expression levels of host genetic factors that modulate heterochromatin. These findings connect the evolution of host genome (genomic TE abundance) with chromatin landscape through the inadvertent harmful effects of the epigenetic silencing of TEs (*Figure 6*). Curiously, the negative association between TE-mediated H3K9me2 enrichment and genomic TE abundance was only observed *within species complex*,

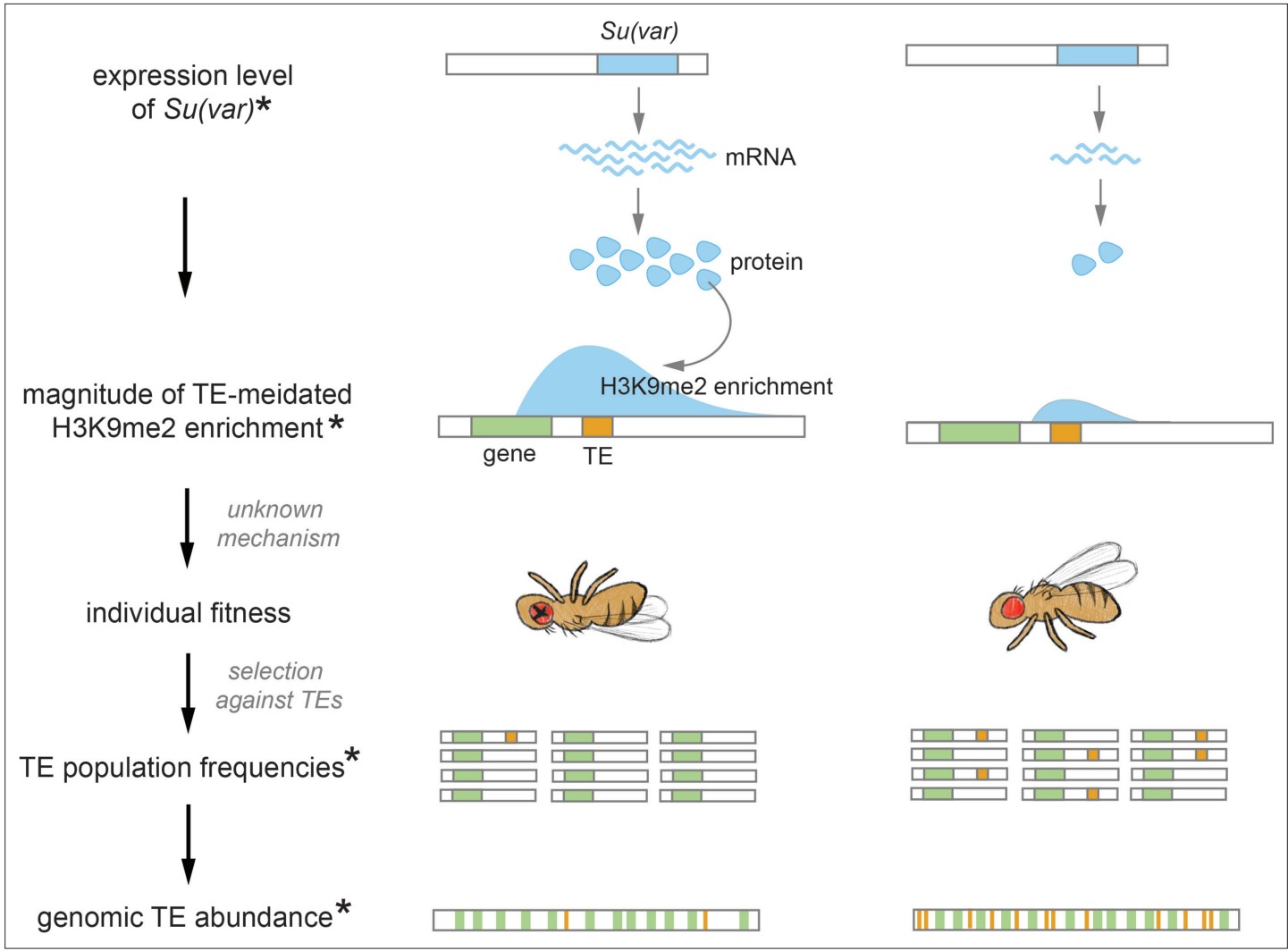

**Figure 6.** Proposed role of the chromatin landscape in determining genomic transposable element (TE) abundance. Our observations suggest that higher expression of *Su(var)s*, which promote a repressive chromatin environment, would result in stronger TE-mediated epigenetic effects (e.g., species on the left). With mechanisms that are yet to be revealed (see Discussion), the stronger epigenetic effects of TEs would reduce individual fitness, resulting in stronger selection against TEs and thus their lower population frequencies. Under the assumptions that the rate of TE increase through transposition is similar across species and the changes in TE copy number is at equilibrium, genomic TE abundance is determined by the strength of selection against TEs. Accordingly, the stronger epigenetic effects of TEs and the associated stronger selection removing them could drive an overall lower genomic TE abundance (e.g., species on the left). Observations made in this study are denoted with *.

pointing that the importance TE-mediated epigenetic effects in determining genomic TE abundance may be on a short evolutionary time scale. Specifically, even though we found stronger epigenetic effects of TEs in species of the *yakuba* complex than those in the *melanogaster* complex, TEs are more abundant in the former. This accumulation of TEs in the genomes of *yakuba* complex is in line with previously observed greater net gains of intron sequences (*Presgraves, 2006*) and duplicated genes (*Hahn et al., 2007*) in *D. yakuba* than those in the lineage leading to *melanogaster* complex, which could suggest a higher tolerance of *D. yakuba* genome with gained sequences. Such lineage-specific mechanism that shapes the gain and loss of sequences over the entire genome may also play an important role in determining genomic TE abundance and potentially override the influence of selection against varying TE-mediated epigenetic effects on a longer evolutionary time scale.

It is worth noting that teasing apart the causality of the observed associations between the epigenetic effects and genomic abundance of TEs is challenging. A reduction in effective population size and accordingly weakened effectiveness of natural selection at removing TEs could drive the accumulation of TEs in some of the species (e.g., *Mérel et al., 2021*). Under this scenario, selection

may favor epigenetic regulation that limits excessive inadvertent spreading of repressive marks from euchromatic TEs, which could also result in our observed weaker TE-mediated epigenetic effects in species with abundant TEs. Moreover, even if the difference in genomic TE abundance is indeed the 'consequence', varying efficacy of selection in purging deleterious TE insertions could also contribute to the observed between-species difference in TE abundance. Specifically, within species complex, estimated nucleotide polymorphism, an indicator for effective population size (*Charlesworth, 2009*), largely follows a similar rank order to our observed magnitude of TE-mediated H3K9me2 enrichment (nucleotide polymorphism, *melanogaster* complex: *D. simulans > D. mauritiana > D. melanogaster* (*Langley et al., 2012*; *Meiklejohn et al., 2018*) and *yakuba* complex: *D. tessieri > D. yakuba > D. santomea* (*Bachtrog et al., 2006*); the only difference in the rank order being the epigenetic effect of TEs is stronger in *D. mauritiana* than in *D. simulans, Figure 5A*). More effective selection at removing TEs in species with larger effective population size (but also stronger epigenetic effects of TEs) is another plausible driver for varying TE abundance that our analysis could not rule out.

Our study focuses on late-stage embryos and would capture the averaged epigenetic effects of TEs across heterogeneous cell types at this developmental stage. If TE-mediated epigenetic effects vary minimally across tissues or cell types, our estimate serves as a good proxy to study the associated fitness consequence. Yet, if TE-mediated enrichment of repressive marks substantially varies between tissues and cell types that have different importance in shaping individual functions, our estimate could fail to capture the most relevant variation in TE-mediated epigenetic effects that determine individual fitness. Even though we did find evidence supporting selection against TE-mediated enrichment of repressive marks that was estimated from whole embryos, future investigation on the variability of such effects across tissues and cell types will be important for the finer dissection of the role of TE-mediated epigenetic effects in individual fitness and, thus, genomic TE abundance.

Proper maintenance of heterochromatin ensures genome function and integrity (reviewed in *Janssen et al., 2018*). Intriguingly, the expression level (see above), copy number (*Levine et al., 2012*; *Ross et al., 2013*; *Helleu and Levine, 2018*), and amino acid sequences (*Levine et al., 2012*; *Sasaki et al., 2019*) of heterochromatin structural and enzymatic components have been observed to vary substantially within and between species. The abundance and composition of heterochromatic repeats also rapidly turnover between genomes (*Larracuente, 2014*; *Wei et al., 2014*; *Wei et al., 2018*). While variation in these genetic factors was postulated to contribute to the evolution of heterochromatin functions (e.g., *Ferree and Barbash, 2009*; *Helleu and Levine, 2018*; *Mills et al., 2019*), our discoveries further extend their evolutionary impacts from the gene-poor genomic dark matter into the gene-rich euchromatin. Heterochromatin modulators may not only determine the local epigenetic impacts of euchromatic TEs on flanking sequences, but also shape TE abundance and, accordingly, the structure and function of the euchromatic genomes. The evolution of euchromatin and heterochromatin, two genomic compartments usually presumed to function independently, may be more interconnected than previously thought by the inadvertent deleterious epigenetic effects of a widespread genetic parasite.

## Materials and methods
### *Drosophila* strains
*Drosophila* strains used in this study include *D. melanogaster* ORw1118 (*Sexton et al., 2012*), *D. simulans* w501 (Drosophila Species Stock Center, strain 1) and Mod6 (strain 2), *D. mauritiana* w12, *D. yakuba* NY73PB (strain 1) and Tai 18E2 (strain 2), *D. santomea* AG01482, and *D. teissieri* GT53W gen14. Flies were cultured on standard medium at 25°C and 12 hr/12 hr light/dark cycles.

### ChIP-seq and RNA-seq experiment
We collected 16–18 hr embryos from each strain to perform ChIP-seq and RNA-seq experiments (with two biological replicates). Before embryo collection, young adults (3–7 days of age) were allowed to lay eggs on fresh apple juice plates for 1 hr at 25°C. We then collected embryos for 2 hr on another fresh apple juice plates and stored those collection plates at 25°C to enrich for 16–18 hr embryos. Chromatin isolation and immunoprecipitation were performed following the modEncode protocol (http://www.modencode.org/) with the following modification. Before splitting the input and IP samples, we added 4 µl of SNAP-ChIP K-MetStat Panel (EpiCypher) to every 10 µg of chromatin.

This concentration allowed ~200–500 barcoded H3K9me2 nucleosome reads with ~15 million Illumina reads, which is the targeted sequencing depth of our samples. The SNAP-ChIP K-MetStat Panel serves as a spike-in and allows normalization between samples. We used H3K9me2 antibody (abcam 1220), which was validated by the modEncode and shows H3K9me2-specific binding (*Egelhofer et al., 2011*), to perform the ChIP experiment. Quantifying ChIP-seq reads corresponding to different histone modifications of the SNAP-ChIP K-MetStat panel (see below) also found this antibody has high specificity to H3K9me2 (*Figure 1—figure supplement 5*). ChIP-seq libraries were prepared using NEBNext Ultra DNA Library Prep Kit for Illumina (NEB) following the manufacturer's protocol. RNAs were extracted using the RNeasy Plus kit (Qiagen) and library prep with Illumina TrueSeq mRNA stranded kit (Illumina) following the manufacturer's protocols. Both ChIP-seq and RNA-seq samples were sequenced on Illumina Hi-Seq4000 with 100 bp, paired-end reads.

## PacBio assemblies

We generated PacBio assemblies for *D. melanogaster* ORw1118 strain with following procedures. High molecular weight DNA was extracted from 450 adults (mixed sexes) using Genomictip Kit (Qiagen) following the manufacturer's protocol. Extracted DNA then underwent SMRTbell library preparation with target insert size as 20 kb and sequenced on one SMRTcell using P6-C4 chemistry on a Pacific Biosciences RSII platform. We used Canu (version 2.0, *Koren et al., 2017*) to assemble a draft genome from the PacBio reads. We then used pbmm2 SMRT Analysis (version 7.0.0) to align the PacBio raw reads to the genome and used Arrow, also from SMRT Analysis, to polish the genome. The final genome has N50=12.1 MB and 178 MB for the total genome size. PacBio assemblies for *D. simulans* and *D. mauritiana* are from *Chakraborty et al., 2021*, and for *D. yakuba* (NCBI: GCA_016746365.2 and GCA_016746335.2), *D. santomea* (NCBI: GCA_016746245.2), and *D. teissieri* (GCA_016746235.2) are from the Andolfatto lab (Columbia University).

## Annotation of TEs in PacBio assemblies

We ran Repeatmodeler2 (version 2.0.1; *Flynn et al., 2020*) on PacBio genome assemblies. NCBI BLASTDB and the analyzed genomes were used as inputs for the BuildDatabase function in Repeatmodeler2. We ran Repeatmodeler2 with options '-engine ncbi -LTRStruct' and selected the output repeats with class as LTR, LINE, DNA, or unknown as potential TE sequences. We required a TE to be at least 500 bp to be included in our analyses.

To assign family identity to identified TEs, we used an iterated blast approach. We first blasted the TE sequences (using blastn; *Camacho et al., 2009*) to TEs annotated in *D. melanogaster* reference genome (version 6.32) and 'canonical' TE sequences of *Drosophila* (retrieved from Flybase September 2019) with following parameters: -evalue 1e-5; -perc_identity 80 for the *melanogaster* complex and -perc_identity 60 for the *yakuba* complex. When an identified TE has blast hits to multiple TE families, we only assigned the TE to a family when at least 80% of the covered query belongs to one and only one family. We then added the annotated TEs to the 'blast database' and repeat the process three more times in order to allow the identification of TEs that are diverged from those in the reference *D. melanogaster* genome or canonical TEs annotated in various *Drosophila* species. TE insertions that are within 500 bp were merged if they are from the same TE family or excluded from the analysis if they belonged to different families. We excluded DINE-1, which are mostly fixed in the *melanogaster* complex species (*Kapitonov and Jurka, 2003*) but underwent a recent burst of activities in *D. yakuba* and likely other closely related species (*Yang and Barbash, 2008*). We also excluded telomeric TEs (HeT-A, TART, and TAHRE), which predominantly locate at the end of chromosomes that are largely heterochromatic. Because most of the TEs included in our blast database are from *D. melanogaster,* it is plausible that our family assignment process is biased against assigning family identity to TEs in the species of the *yakuba* complex. Accordingly, except for analyses that require TE family identity (e.g., *Figure 2B*, *Figure 2C*, *Figure 2—figure supplements 1 and 2* and *Figure 2—figure supplement 4*, and the estimation of TE population frequencies, see below), we included all TEs that are at least 500 bp in the analysis, irrespective whether we could assign their family identities or not. To identify whether TEs were full-length or truncated, TE sequences were aligned to the annotated *D. melanogaster* canonical sequences for the respective families using MAFFT (v7.505, Katoh and Standley 2013). TEs whose length is at least 70% of the annotated *D. melanogaster* canonical sequences were

considered intact while others were deemed truncated. This *D. melanogaster*-centric analysis may bias against assigning TEs in other species as full length.

## Identification of euchromatin/heterochromatin boundaries

Because we are interested in the epigenetic effects of euchromatic TEs, our analysis excluded those that are in or close to heterochromatin. To determine the euchromatin-heterochromatin boundaries in each genome, we generated H3K9me2 fold-enrichment tracks using MACS v2.7.15 (*Zhang et al., 2008*). We then visualized the H3K9me2 enrichment genome-wide using IGV (version 2.10.2, *Thorvaldsdóttir et al., 2013*) to identify the sharp transition in H3K9me2 enrichment and used positions that are 0.5 Mb 'inward' (toward the euchromatin) from the transition as the euchromatin-heterochromatin boundaries. These conservative euchromatin-heterochromatin boundaries are expected to minimize the influence of constitutive heterochromatin in influencing our analysis.

## ChIP-seq analysis

In order to align reads originated from the spike-in control (SNAP-ChIP K-MetStat Panel, Epicypher), we combined PacBio genomes with DNA sequence barcode of the SNAP-ChIP K-MetStat Panel, which were serve as the 'reference genome' for Illumina sequence alignment. Raw reads were trimmed using *Trimmomatic* v0.35 (*Bolger et al., 2014*) before aligning to the reference genome with bwa mem v 0.7.16a (*Li and Durbin, 2009*). Read with low mapping quality (q<30) and reads that mapped to multiple locations were removed using Samtools (*Li, 2011*). We used the abundance of reads corresponding to the spike-in H3K9me2 nucleosome to normalize the background level of H3K9me2, following *Lam et al., 2019*. Briefly, we calculated the enrichment of spike-in H3K9me2 nucleosome reads as: $E_{si}$ = (barcode fragments in ChIP)/(barcode fragments in input). We then used bedtools v2.25.0 (*Quinlan and Hall, 2010*) to obtain read coverage across the genome for ChIP and input samples. For 25 bp nonoverlapping windows across the genome, we calculated the per-locus enrichment as $E_{locus}$ = (fragment coverage in ChIP)/(fragment coverage in input). The HMD for H3K9me2 in a particular 25 bp window was then estimated as $E_{locus}/E_{si}$. Windows with fewer than five fragment coverage in input samples were treated as missing data.

## Estimation of TE-mediated H3K9me2 enrichment

For each TE, we normalized the local H3K9me2 HMD level in its flanking sequence by the median HMD for regions 20–40 kb upstream and downstream of each TE, following *Lee and Karpen, 2017*. The reasons behind this approach come from the observations that TE-mediated spreading of repressive epigenetic marks is usually within 10 kb in *Drosophila* (*Lee, 2015*; *Lee and Karpen, 2017*). We then divided the 20 kb upstream and downstream from a TE into 1 kb nonoverlapping windows and, for each window, calculated the median of normalized H3K9me2 HMD among its 40 25bp-HMD units (see above, m-HMD); at least 10 HMD estimates are required to calculate m-HMD for a 1 kb window. To estimate the magnitude of TE-mediated local enrichment of H3K9me2, we calculated the m-HMD for the 1 kb left and right flanking regions separately and took the average of the two sides. To estimate the extent of H3K9me2 spreading, we examined whether m-HMD is above one, which indicates that the H3K9me2 HMD level for the window is higher than that of the local background. We scanned across windows, starting from those right next to TEs and identified the farthest window in which the m-HMD was consecutively above one. We then used the average for the two sides as the extent of H3K9me2 spreading. The estimates for HMD magnitude or extent from two replicates positively correlate (*Spearman rank correlation coefficient* $\rho$ =0.19–0.84, p<10^{-8}, *Figure 1—figure supplement 6*). Curiously, the strength of correlation for the estimated magnitude or extent of TE-mediated epigenetic effects among replicates is low for some samples (e.g., magnitude of the effect for *D. melanogaster*). We performed IDR (irreproducible rate) analysis between replicates (*Li et al., 2011*) and found limited associations between IDR and the strength of between-replicate correlation (*Figure 1—figure supplement 7*). For instance, *D. melanogaster* shows the lowest correlation for the magnitude of TE-mediated H3K9me2 enrichment between replicates, but has decent consistency between replicates by IDR analysis. A plausible cause for this discrepancy is that TE-mediated enrichment of repressive marks does not have the typical characteristic of 'enrichment peaks' (*Lee and Karpen, 2017*; *Lee et al., 2020*) and thus could be oftentimes missed by custom pipelines that were designed to identify 'sharp peaks' (reviewed in *Park, 2009*; *Nakato and Shirahige, 2017*). For

instance, only 12% of TEs showing local enrichment of H3K9me2 was detected by peak calling in a previous study (*Lee et al., 2020*). On the other hand, IDR analysis estimates the reproducibility of called peaks between replicates and might not include the majorities of regions with TE-mediated enrichment of repressive marks. More importantly, the spreading of repressive marks from constitutive heterochromatin is a variegated phenotype and was observed to vary between cells and individuals (reviewed in *Elgin and Reuter, 2013*), which could explain the variability between replicates. While the rank order of significance for called peaks are mostly consistent between replicates and falls along the diagonal line for significant called peaks for most samples (*Figure 1—figure supplement 7*), we noticed that this trend in *D. simulans* strain 2 is weak. We thus tried excluding this sample in analysis that tested the associations between genomic TE abundance and TE-mediated epigenetic effects (*Figure 5A*) and reached similar conclusion (see text).

To proceed with following analysis, we averaged the estimates from two replicates to generate the magnitude and extent of H3K9me2 enrichment. It is worth noting that the estimated extent of H3K9me2 spreading with different thresholds of m-HMD strongly correlate (*Spearman rank correlation coefficient* $\rho$ =0.69–0.87, p<10$^{-16}$, *Figure 1—figure supplement 8*), suggesting the robustness of such estimate. In the analysis, a TE was considered showing 'epigenetic effects' when the magnitude of H3K9me2 enrichment is above one or has at least 1 kb spreading of H3K9me2 (see text). Because the magnitude and extent of H3K9me2 enrichment do not follow normal distribution even after log transformation (Anderson-Darling normality test, for all test p<2.2e$^{-16}$), we chose to perform nonparametric *Spearman rank correlation tests* when investigating the associations between TE-mediated epigenetic effects and factors of interests. The nonparametric *Spearman rank correlation tests* are also less sensitive to the effects of outliers, which is critical for our goal of identifying genome-wide patterns.

For two species (*D. simulans* and *D. yakuba*), we also investigated the epigenetic effects of TEs by comparing the enrichment of H3K9me2 at homologous sequences with and without TE insertions and contrast that to single species-based method (see above). We first used minimap v2.17 (*Li, 2018*) to generate assembly-to-assembly alignments of the PacBio genomes and identify homologous sequences for the two strains of a species (paftools.js from minimap). We ran MACS v2.7.15 (*Zhang et al., 2008*) to identify shared peaks of H3K9me2 enrichment using liberal significant threshold (with broad-cutoff p=0.5). TEs in these shared peaks were then excluded from the analysis because we could not determine if the enrichment of H3K9me2 for these regions were induced by TEs. We also excluded focal TEs whose homologous sequences were within 1 kb of another TE in the alternative strain. The magnitude of H3K9me2 enrichment was estimated as the m-HMD of the focal TE in the focal strain standardized by the m-HMD in the 1 kb TE-flanking regions in the homologous sequence in the alternative strain. The extent of H3K9me2 spreading was measured as the distance for the farthest windows from TEs that the m-HMD was consecutively higher in focal strain than that in the alternative strain. The estimates based on one or two genomes significantly correlate (*Figure 1—figure supplement 1*; see text).

## Inference of genic H3K9me2 enrichment

Reference genome sequences and annotations for *D. melanogaster, D. simulans, D. mauritiana,* and *D. yakuba* were downloaded from NCBI Datasets (*D. melanogaster*, version Release 6 plus ISO1 MT; *D. simulans*, version ASM75419v2; *D. mauritiana*, version ASM438214v1; *D. yakuba*, version dyak_caf1). The annotations were lifted to PacBio assemblies by aligning the PacBio assemblies to NCBI references using minimap v2.17 (*Li, 2018*). Because the annotations for *D. santomea* and *D. teissieri* were not available when we performed the analyses, we used MAKER v2.31.8 (*Holt and Yandell, 2011*, p. 2) to annotate the PacBio assemblies. Specifically, we used Trinity-v2.85 (*Grabherr et al., 2011*) to de novo assemble the mRNA-seq for the genome as EST evidence for MAKER. We also supplied the protein sequences from *D. melanogaster, D. sechellia, D. simulans, D. mauritiana, D. erecta,* and *D. yakuba* as protein homology evidence for MAKER. We then ran MAKER with the default settings to obtain annotations for *D. santomea* and *D. teissieri*. To study the relationships between the epigenetic effects of TEs and the epigenetic states of neighboring genes, we assigned each TE to its closest gene and distinguished whether it inserted at the 5' or 3' side of the gene. Genic H3K9me2 enrichment is estimated as the average of HMD of the gene body, excluding genes with TEs inserted. When comparing the H3K9me2 enrichment level of homologous genic alleles with and without adjacent TEs,

we calculated z-score as: (mean HMD of allele with nearby TE – mean HMD of allele without nearby TE in the alternative strain)/(standard deviation of both strains). Because genic H3K9me2 enrichment does not follow normal distribution, even after log transformation (Anderson-Darling normality test, for all test $p<2.2e^{-16}$), we chose to perform nonparametric *Spearman rank correlation tests* when investigating the associations between genic HMD and factors of interests.

## Association between TE-mediated epigenetic effects and TE abundance across species

To examine whether the epigenetic effects of TEs associate with TE abundance across species while accounting for species complex effects, we performed linear regression analyses using the following model: TE abundance ~ species complex (*melanogaster* or *yakuba* complex) + epigenetic effect. ANOVA F-test was used to examine whether the epigenetic effect was significant. We also performed PGLS (*Grafen, 1989*; *Martins and Hansen, 1997*) analysis using a tree adapted from *Turissini and Matute, 2017*; *Chakraborty et al., 2019*, with arbitrary branch lengths: ((((Dsim_strain1:0.1,Dsim_strain2:0.1):0.15, Dmau:0.25):3,Dmel:3.25):7.25, (((Dyak_strain1:0.1,Dyak_strain2:0.1):0.9,Dsan:1):1.75, Dtei:2.75):7.75). Although the significance level was sensitive to the within-species branch lengths, the sign of the coefficients remained unchanged. To fit the above model, we used gls function from nlme package with a Brownian correlation structure based on the tree (imported using ape package) in R.

## Gene expression analysis

To estimate gene expression abundance, we mapped the raw RNAseq reads to the annotated genomes with STAR v2.6.0 (*Dobin et al., 2013*) with options `--quantMode` TranscriptomeSAM Gene-Counts `--chimFilter` None. In order to compare the expression levels between different species, we used reciprocal best blasts between *D. melanogaster* and one other species to identify one-to-one orthologs using blastn (version 2.8.1). We then obtained a set of shared orthologs among six species to compare expression levels. For this set of shared orthologs, we estimated the RPKM (reads per kilobase per million reads) as the averaged RPKMs from two replicates (RPKMs from two replicates strongly correlate; *Spearman rank correlation coefficient* $\rho$ >0.98 for all genomes, $p<10^{-22}$). Genes with 0 RPKM in both replicates were excluded from the analysis. We then ranked genes from the highest to lowest RPKMs in each strain to get expression rank.

## Estimation of TE population frequencies

Raw Illumina reads for *D. melanogaster* (*Lack et al., 2015*), *D. simulans* (*Rogers et al., 2014*), *D. mauritiana* (*Garrigan et al., 2012*), and *D. yakuba* (*Rogers et al., 2014*) were downloaded from SRA (SRP006733, SRP040290, SRP012053, and SRP029453, respectively). We used FastQC v0.11.7 (https://qubeshub.org/resources/fastqc) to check the read quality and TrimGalore v0.6.0 (*Babraham Bioinformatics, 2022*) to remove adapter and low-quality sequences. Illumina reads were then mapped to the corresponding PacBio assembly using bwa mem v0.7.16a (pair-end mode). We further used samtools to filter out reads with mapping quality smaller than 50 (MAPQ < 50).

To call the presence/absence of TEs, we followed the basic ideas developed in *Cridland et al., 2013*; *Lee and Karpen, 2017*, with following modifications. Briefly, we parsed out reads that uniquely mapped to the ±500 bp around TEs annotated in the PacBio assembly using *seqtk v1.3-r107-dirty* (https://github.com/lh3/seqtk; *Li, 2022*). Parsed reads were assembled into contigs using *phrap v1.090518* (*Ewing and Green, 1998*) with parameters from *Cridland et al., 2013*. We mapped the resultant contig to PacBio assembly using bwa mem (default options). If a *single* contig mapped across 20 bp upstream and downstream of an annotated TE, a TE is called absent. To determine whether a TE is present, we evaluated whether *two* separate contigs spanned across the start and end of a TE's boundaries, respectively, with at least 30 bp inside the TE and 20 bp outside the TE and a minimum total alignment length of 50 bp. We also evaluated contigs that mapped within annotated TEs. A TE is called present if there are two contigs spanning both the start/end of the TE insertion respectively *or* there is one contig spanning either the start or end of the TE and one contig aligned within TE insertion. A TE is considered missing data if none of the above criteria were met.

In the initial runs, we noticed that the failure to identify some TEs in the tested strain is due to the imprecise TE boundaries annotated by RepeatModeler2. In order to reduce the rates of missing data,

we refined the called TE boundaries with following procedures. TE absence calls could be viewed as deletion structural variants (SVs) with respect to the PacBio genome. We thus used *Lumpy-sv v0.2.13* (*Layer et al., 2014*) to call SVs in each strain. Deletions that overlap with the annotated TEs were extracted using sytyper v0.0.4 (*Chiang et al., 2015*) and merged using svtools v0.5.1 (*Larson et al., 2019*) to refine TE boundaries, which were then used in the above pipeline. For most TEs, the differences between the updated and RepeatModeler2 boundaries are short and only 5.2–7.9% of TEs boundaries in each genome were significantly updated (≥20 bp from original boundaries). Code for the TE calling pipeline can be found at https://github.com/harsh-shukla/TE_freq_analysis, (*Huang, 2022* copy archived at swh:1:rev:24218fab83996f657e489402c5ff1c4cc06bfe9c).

## Quantification of the heterochromatic repeats

To identify the repeat sequences enriched in the heterochromatic regions of the genome, we first used KMC (version 3.1.1, *Kokot et al., 2017*) to quantify the 12-mers in our H3K9me2 IP and matching input samples. In order to compare 12-mers abundance between IP and input libraries, we normalized the 12-mers counts by the number of reads mapped uniquely to the PacBio genome with at least 30 mapping quality score. 12-mers that have at least a threefold enrichment in an IP sample when compared to its matching input sample were considered as heterochromatic repeats.

Because PCR amplification of sequencing libraries were shown to influence the quantification of simple repeats (*Wei et al., 2018*), we sequenced the genomes of focused strains with Illumina PCR-free library preparation. We extracted DNA with 40 females using DNeasy Blood & Tissue Kit (Qiagen), following the manufacturer's protocol. Extracted DNA was then prepared into Illumina sequencing libraries with PCR-free protocol and sequenced with 150pb paired-end reads by Novogene (Sacramento, CA). We then ran KMC3 on these libraries to quantify the abundance of heterochromatic repeats identified above. In order to compare across strains, the number of heterochromatic repeats were further normalized with the total number of reads from either the orthologous region.

## *Drosophila* mutant crosses and eye pigmentation assay

To investigate the mutant effects of identified candidate *Su(var)s*, we followed approaches outlined in *Sentmanat and Elgin, 2012*. Specifically, we used a strain developed in *Sentmanat and Elgin, 2012*, which has a TE *1360* placed next to *mini-white in y; w* background. It was found that 1*360*-mediated epigenetic effects influence the expression level of *mini-white* and thus the intensity of eye pigmentation. Our analysis only tested the effects of *Su(var)s* whose existing mutants do not have any eye markers to avoid confounding effects on the quantification of eye pigmentation. We crossed 3- to 5-day-old virgin females of this strain to males of the *Su(var)* mutant strains or a control strain and then quantified the eye pigmentation level in 25–30 three- to five-day-old F1 males following methods described in *Sentmanat and Elgin, 2012*. Three independent crosses were performed for each mutant. Strains used in the analysis include BDSC 6398 (*Hsc70-4* mutant), BDSC 11537 (*RpLP0* mutant), BDSC 30640 (*Bin 3* mutant), BDSC 36511 (*Ago 2* mutant), and BDSC 6559 (*y; w*, control).

## Acknowledgements

We greatly appreciate JJ Emerson and Peter Andolfatto for generously providing the PacBio genomes and corresponding *Drosophila* strains, Sally Elgin for providing 1360-mini-white strains, Giacomo Cavalli for sharing ORw1118 line, Bloomington Drosophila Stock Center for providing *Su(var)* mutants, and David Acevedo and Jasmine Osei-Enin for technical assistance. Patrick Reilly and Mahul Chakraborty provided helpful guidance on the PacBio genome assembly, and Kevin Brick for discussion about ChIP-seq normalization. We thank University of California High-Throughput Genomics Facility and High Performance Cluster at UC Irvine for sequencing and computational resources. We also appreciate Aneil Agrawal, Ching-Ho Chang, Jae Choi, Brandon Gaut, Mia Levine, Aline Muyle, and members of the Lee lab for providing helpful discussions and comments on the manuscript and Leila Lin for assistance in drawing the model figure. We also thank three reviewers for their helpful comments. This work was supported by NIH R00GM121868 and R35GM14292 to YCGL.

## Additional information

### Funding

| Funder | Grant reference number | Author |
|---|---|---|
| National Institutes of Health | R00GM121868 | Yuh Chwen G Lee |
| National Institutes of Health | R35GM14292 | Yuh Chwen G Lee |

The funders had no role in study design, data collection and interpretation, or the decision to submit the work for publication.

### Author contributions

Yuheng Huang, Data curation, Formal analysis, Validation, Investigation, Methodology, Writing - original draft, Writing - review and editing; Harsh Shukla, Software, Writing - review and editing; Yuh Chwen G Lee, Conceptualization, Resources, Data curation, Formal analysis, Supervision, Funding acquisition, Investigation, Visualization, Methodology, Writing - original draft, Project administration, Writing - review and editing

### Author ORCIDs

Yuheng Huang (ID) http://orcid.org/0000-0002-1586-953X
Yuh Chwen G Lee (ID) http://orcid.org/0000-0002-0081-7892

### Decision letter and Author response

Decision letter https://doi.org/10.7554/eLife.81567.sa1
Author response https://doi.org/10.7554/eLife.81567.sa2

## Additional files

### Supplementary files

• Supplementary file 1. Correlation coefficient and regression coefficient for the associations between *Su(var)* expression rank and the magnitude of transposable element (TE)-mediated H3K9me2 enrichment. Genome-wide percentiles for correlation coefficient: –0.7142 (5%) and –0.5952 (10%); genome-wide percentiles for regression coefficient: –0.4208 (5%) and –0.3208 (10%).

• MDAR checklist

### Data availability

Short-read data for ChIP-seq, RNA-seq, and Illumina whole-genome sequencing have been deposited to NCBI BioProject under accession number PRJNA855483. PacBio long-read data and genome assembly of *D. melanogaster* ORw1118 strain have been deposited to NCBI BioProject under accession number PRJNA855235.

The following datasets were generated:

| Author(s) | Year | Dataset title | Dataset URL | Database and Identifier |
|---|---|---|---|---|
| Lee YCG | 2022 | Evolutionary causes and consequences of transposable elements' epigenetic effects | https://www.ncbi.nlm.nih.gov/bioproject/?term=PRJNA855235 | NCBI BioProject, PRJNA855235 |
| Lee YCG | 2022 | Evolutionary causes and consequences of transponsable elements' epigenetic effects | https://www.ncbi.nlm.nih.gov/bioproject/PRJNA855483 | NCBI BioProject, PRJNA855483 |

The following previously published datasets were used:

| Author(s) | Year | Dataset title | Dataset URL | Database and Identifier |
|---|---|---|---|---|
| Lack JB | 2015 | "1000" *Drosophila* Genomes - DPGP3 | https://trace.ncbi.nlm.nih.gov/Traces/index.html?view=study&acc=SRP006733 | NCBI, SRP006733 |
| Rogers RL | 2014 | *Drosophila* simulans strain:multiple Genome sequencing | https://trace.ncbi.nlm.nih.gov/Traces/index.html?view=study&acc=SRP040290 | NCBI, SRP040290 |
| Rogers RL | 2014 | *Drosophila* yakuba strain:multiple Genome sequencing | https://trace.ncbi.nlm.nih.gov/Traces/index.html?view=study&acc=SRP029453 | NCBI, SRP029453 |
| Garrigan D, Kingan SB, Geneva AJ, Vedanayagam JP, Presgraves DC | 2014 | *Drosophila mauritiana* genome sequencing | https://trace.ncbi.nlm.nih.gov/Traces/index.html?view=study&acc=SRP012053 | NCBI, SRP012053 |

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
