## [Editor Report]

Transposable elements are genomic parasites and the fraction of the genome that is made up of such elements varies greatly between species, and models suggest that this must reflect the balance between the rate at which they multiply, and the rate at which selection purges them from the genome. Precisely how selection acts against transposable element insertions is not clear. This paper provides evidence that the strength of selection depends on the extent to which epigenetic silencing spreads to nearby genes – although the mechanism is obscure, as gene expression is not affected. This is a very interesting hypothesis that deserves more attention, and the paper is an excellent example of trying to combine population genetics models with a mechanistic understanding of the process modeled.

---

## [Decision Letter]

[Editors' note: this paper was reviewed by Review Commons.]

---

## [Author Response]

**1. General Statements**

This study builds upon a previous study published in *eLife* (Lee and Karpen 2017), which established the importance of transposable element (TE)-mediated local enrichment of repressive marks in TE evolution in the model species, *D. melanogaster.* In the current study, we investigated this TE-mediated phenomenon in six closely related *Drosophila* species to have a good phylogenetic resolution, aiming to address step-by-step how such TE-mediated effects may shape the evolution of genomic TE content, a long-standing question in evolutionary genomics. Our analyses successfully connect species-specific host chromatin regulation, TE-mediated epigenetic effects, the strength of natural selection against TEs, and genomic TE abundance unique to individual species, while uncovering the evolutionary causes for the wide variety of TE profiles between species.

We appreciate the reviewers for their overall positive response and helpful comments on the previous versions of the manuscript. We appreciate the opportunity to correct our previous oversights, revise analysis to corroborate findings, and improve interpretations of our observations. Thanks for these suggestions, we believe the revised manuscript is much improved and hope it is now acceptable for publication in *eLife*. Please see below for our pointby-point responses to reviewers’ comments.

In addition to analyses suggested by the reviewers, we also updated one of our analyses to avoid non-independence of samples. Specifically, in Figures 3A and B and Supplementary Figures S1-6, we investigated the associations between TE-mediated epigenetic effects and genic epigenetic states/expression. In our previous analysis, a TE could be associated with more than one gene, if it is the nearest TE to more than one gene. Yet, this could lead to nonindependence of observations. In this revised manuscript, we restricted the analysis to TE and its nearest gene, excluding this potential issue. The results are qualitatively the same and quantitatively similar, and our conclusions from these results remain unchanged.

**2. Point-by-point description of the revisions**

Reviewer #1 (Evidence, reproducibility and clarity (Required)):Summary:In this work titled "Species-specific chromatin landscape determines how transposable elements shape genome evolution", Huang, Shukla and Lee examined the epigenetic effect of euchromatic TE insertions in 6 *Drosophila* species (3 in melanogaster species complex and 3 in yakuba species complex). They profiled the H3K9me2 mark and transcriptome from 16-18hr embryos and, using genomics, population genetics and statistical tools, they extended claims made previously in D. mel to 5 more species: (1) H3K9me2 spreads to nearby regions, exerting an epigenetic effect; (2) TEs with epigenetic effects are selected against. More importantly, this work aims to identify rules that explain the differences in TE-mediated epigenetic effects across the 6 species examined and how their distinct chromatin landscape might affect the way in which TE-mediated epigenetic effects shape genome evolution. The analysis seems to be well done, rather rigorous, and considers important counterarguments in several places throughout the manuscript. The figure is clear, and the method is well detailed. I have some comments that should be addressed before publication, involving additional bioinformatic analysis and text edits to either tone down or re-frame the conclusions.Major comments:1. Exclusive usage of Spearman rank correlation for key conclusions requires further support. I worry that using Spearman rank correlation loses too much of the quantitative information in e.g., gene expression. Have authors ever looked at Pearson's correlation (perhaps upon log-transformed levels when necessary) and does it lead to similar or different results? While this concern holds for all conclusions based on Spearman rank correlation, it is particularly a concern for analysis involving gene expression, especially for the surprising claim that TE-mediated K9 spreading does not affect adjacent gene expression.

We agree with the reviewer that, compared to parametric tests (e.g., *Pearson tests*), nonparametric tests have more restricted statistical power. An insignificant *Spearman correlation test* could either be due to true biological reasons or simply a lack of statistical power. Yet, we found that the magnitude and extent of TE-mediated H3K9me2 enrichment and the genic HMD do not follow normal distribution even after log transformation (Anderson-Darling normality test, all p-value < 10^-16^). Accordingly, we are unable to perform the suggested *Pearson* correlation tests. Another reason why we would like to stick with the *Spearman* tests is because it is less sensitive to outliers than the *Pearson correlation tests.* This is especially important because our goal is to identify genome-wide average patterns.

It is worth noting that, in addition to *Spearman rank* correlation analysis, we also compared the expression level between homologous alleles with and without TEs showing H3K9me2 spreading in *D. simulans* and *D. yakuba* (Figure 3C and Figure 3 – Supplementary Figure S7-9) and found consistent results. We included additional discussion to explain why we chose to use nonparametric tests (p50, line 12-18; p51, line 27-31) and emphasize that, in addition to the correlation analysis, other tests also support limited impacts of TE-mediated H3K9me2 enrichment on gene expression (p39, line 13-15).

We agree with the reviewer that the observation of the limited associations between TE-mediated H3K9me2 spreading and gene expression is surprising. We hope our addressing the reviewer’s comment 3 and additional discussions and citations of other relevant/suggestive studies that reached similar conclusions (p39, line 15-22) help address this issue.

2. Choice of binning procedure over correlation analysis seems unjustified. I understand the binary binning in e.g., Figure 1C, but for quantitative variables e.g., distance to TE/gene, allele frequency in Figures3 and 4, I worry that the binning procedure based on somewhat arbitrary cutoffs could mask the real relationship. Are the conclusions robust against the cutoff chosen?Is there actual correlation between the two variables examined?

For previous Figure 3A and 3B (now Figure 3 – Supplementary Figures 1-4), following the reviewer’s suggestion, we performed regression analysis that uses the distance between genes and TEs as continuous variables to test whether the associations between TE-mediated epigenetic effects and genic H3K9me2 enrichment/expression depends on TE-gene distance. Such analysis reached the same conclusion and is now included in p15, line31 – p16, line 4; p16, line 16-19. We chose to present the analyses that bin genes based on their distance to TEs because it is challenging to interpret them as three-dimensional graphs; current Figure 3 – Supplementary Figures 1-4 have TE-mediated epigenetic effects as X-axis, genic attribute as Y-axis, and bins of gene-TE distance as a binary color. It is worth noting that, in this revised version of the manuscript, we chose to present the correlation coefficients between TE-mediated H3K9me2 enrichment and genic epigenetic states/expression (new Figures 3A and 3B). This is because these figures provide a good visual summary for all the comparisons (magnitude/extent, genic H3K9me2 enrichment/expression, and TEs 5’ and 3’ to genes) and are easier to be interpreted than original Figures 3A and 3B (now Figure 3 – Supplementary Figures 1-4).

In Figure 3D, we estimated the z-score to compare the extent of TE-mediated spreading of H3K9me2 on the genic and intergenic sides between TEs close and distant from genes (the binning is based on 50% percentile of the distance between TEs and genes). Following the reviewer’s suggestion, we performed correlation tests between the z-score and TE-gene distance and found consistent results – we identified significant negative associations between the z-score of a TE and its distance from the nearest gene. These results are now presented in p19, line18-23.

For the population frequencies analysis (Figure 4), we were unclear about how we categorized low-frequency and high-frequency TEs. A TE is “low-frequency” if it is only identified in the focused strain, but not in the population (in other words, a private TE insertion). This information is now included in p23, line 19-21. Surveys on TE population frequencies in *Drosophila* found highly skewed frequency spectra and that the majority of the TEs appear as singletons (i.e., present in only one individual in the population sample); analysis typically categorizes TEs according to whether they are singleton (low-frequency in our analysis) or not (high-frequency in our analysis). Accordingly, we stick with this binning in our analysis.

3.1. TE position relative to genes seems oversimplified. The manuscript compares TEs that are close to or far from genes, but what happens to TEs in the genes (either in UTRs, CDSs or introns)? Do these different TE positions inside and outside genes exert different effects on K9 magnitude, K9 spreading extent, gene expression and/or allele frequency in the population?

(We split the reviewer’s comment #3 into two and answered them individually).

Following the reviewer’s suggestion, we compared the magnitude and extent of TE-mediated H3K9me2 enrichment between intergenic, intronic, and exonic TEs. While we don’t find any consistent significant differences for the *magnitude* of TE-mediated epigenetic effects, intergenic TEs exert a larger *extent* of H3K9me2 enrichment than other TEs in multiple genomes. These results are now included in p13, line 4-15 and Figure 2 – Supplementary Figure S4.

Our goal is to identify how TE-mediated epigenetic effects associate with genic epigenetic status, gene expression, and TE population frequencies. The difference in TE-mediated epigenetic effects between intergenic and genic TEs would have been captured by the estimated magnitude and extent of H3K9me2 enrichment. Accordingly, for the downstream analysis, we did not further distinguish between genic and intergenic TEs with one exception. Because TEs inside genes could influence gene expression through mechanisms other than altering the epigenetic states, a major confounding factor of previous studies (reviewed in (Choi and Lee 2020 PLoS Genetics, Kelleher et al. 2020 Trends in Genetics)), we excluded TEs inside genes when testing the associations between TE-mediated epigenetic effects and gene expression. We further emphasized this point in p15, line 15-18.

3.2. Are there any differences to TEs being 5' or 3' to the genes, since TEs 5' to the genes are more likely to affect the promoter? I worry that simple comparison of close vs. far might skew the actual relationships.

We appreciate the reviewer for pointing this out. Our additional analysis following the reviewer’s suggestions led to some new observations that further corroborate our conclusions, mainly whether neighboring gene expression could influence the extent of TE-mediated epigenetic effects.

We agree with the reviewer that TE-mediated epigenetic effects on the epigenetic states and expression of genes could differ when TEs are 5’ or 3’ to genes, especially given that TEs 5’ to genes should be more likely to influence the promoter. We performed correlation analyses between TE’s epigenetic effects and genic H3K9me2 enrichment or expression separately for TEs 5’ and 3’ to genes and reached similar conclusions to those of analyzing all TEs together (Figure 3B). Specifically, we found significant associations between TE’s epigenetic effects and genic H3K9me2 enrichment for both TEs 5’ and 3’ to genes, such associations depend on TE-gene distance, and we did not notice any major difference between TEs 5’ and 3’ to genes. On the other hand, we observed limited associations between TE-mediated epigenetic effects and gene expression, except for one incidence. These results are included in p16, line 23-p15, line 16 and Figure 3B.

We also tested whether the impact of adjacent gene expression on the extent of TE-mediated enrichment of H3K9me2 differs between TEs 5’ or 3’ to genes. We found that the difference in the extent of TE-mediated H3K9me2 enrichment is greater for TEs 5’ than 3’ to genes, likely due to the effects mentioned by the reviewer and providing stronger support for the hypothesized effects of gene expression on the extent of TE-mediated local H3K9me2 enrichment. We included these results in p20, line 5-10, Figure 3 – Supplementary Figure S12.

4. Transcriptional activity of TEs was not considered. Many TEs are (5'-) truncated and thus transcriptionally inactive. There is literature that suggests K9 installation and spreading only happen on transcriptionally active TEs. Have authors considered separating active/inactive or intact/truncated TEs? Combining TEs with different transcriptional activity (or K9 biology) despite the same length, family or position could confound the analysis and lead to different conclusions.

We agree with the reviewer that this is an important axis of the analysis that we did not consider. We categorized TEs whose length is shorter than 70% of the annotated, canonical sequence in *D. melanogaster* as truncated, and otherwise as full length. Consistent with what is suggested by the reviewer, full-length TEs tend to show a larger magnitude and extent of H3K9me2 enrichment. These results are now included in p11, line 18-26 and Figure 1 – Supplementary Figure S4. Because our following analysis focuses on how TE-mediated epigenetic effects, instead of individual TE attributes, influence genic H3K9me2

enrichment/expression and selection, we did not further consider full-length/truncated TEs for the downstream analysis.

5. Usage of 16-18hr whole-embryo averaged transcriptome and H3K9me2 epigenome require justification. The 16-18hr embryo is presumably consisted of a very heterogenous population of cells, how does such a population average reflect bona fide biology? Authors should either justify this or comment explicitly in text on potential problems such datasets could have on major conclusions.

We agree with the reviewer that 16-18hr embryos consist of heterogeneous cell types, which are precursors of different tissues and organs of a whole animal. By examining the epigenetic effects of TEs in whole embryos, our analysis would capture the averaged effects across cell types. If the relative importance of tissues/organs in determining individual fitness is proportional to the abundance of their specific cell precursors, our estimate could provide a reasonable estimate of the epigenetic effects of TEs that may be relevant to individual performance and thus fitness. Yet, if the fitness of an individual is predominantly determined by a tissue/organ that originated from a rare cell type and TE-mediated epigenetic effects vary substantially between cell types, our analysis of whole embryos may not be a good indication of the associated fitness consequence. Even though we did find evidence supporting selection against TE-mediated enrichment of repressive marks estimated from whole embryos, future investigation on the variability of such effects across tissues and cell types, which we aim to address in the future, will be important to further dissect the role of TE-mediated epigenetic effects in individual fitness and, thus, genomic TE abundance.

Following the reviewer’s suggestion, we emphasized that our dataset should be viewed as an average of all cell types and stated the potential caveats in p42, line 22-p43, lin2.

6. Euchromatic TE and genome-wide TE abundance should not be used interchangeably. This work is done solely on euchromatic TEs, but throughout the text, authors comment on how a factor examined might shape genome-wide TE abundance or even genome evolution. I wonder how much euchromatic TE copy number reflects genome-wide TE abundance? I'd expect heterochromatin harbors most TE copies and to be the major place that determines the overall TE abundance. Authors should either discuss how euchromatic TEs do reflect genome-wide TE abundance (and genome evolution) or change all related claims to better describe analysis done on euchromatic TEs.

We agree with the reviewer that this is a very important point that we did not specify previously. The genome-wide TE abundance we discussed in the text and also in the field mainly concerns euchromatic TEs. This is because of two major reasons. One is the technical difficulties associated with assembling and annotating TEs in the highly repetitive heterochromatic regions of the genome. The other reason is that, in the few species where heterochromatin has been assembled, most TEs in the heterochromatic regions are fragmented and have lost of transposition ability, making them no longer important players in the evolutionary dynamics of TEs (i.e., TE number increase through transposition and decrease through selection and excision). Following the reviewer's suggestions, we clearly defined that our study focuses on euchromatic TEs in the opening paragraph (p2, line 9-30), in the first paragraph of result (p7, line 21-22), and in the opening paragraph of discussion (p37, line 7-16).

We also emphasized that our study focused on euchromatic TEs and revised our claim at places throughout the manuscript (e.g., p6, line 8; p29, line 8, p37, line18).

7. Potential contradiction between key conclusion and the "species complex effect". In Figures5A and 5C, the species complex effect is obvious and, when we compare the two species complexes, one would conclude the opposite of what the authors attempted to argue. For instance, K9 and TE copy number correlate between mel and yak species complexes. Does this actually mean that there is different biology (or forces) at play at different evolutionary timescales? Authors' major claims seem valid within a shorter period of time (<3 million years) within each of the two species complexes, but they do not appear to hold when one looks at longer timescales. If so, the authors should modify key conclusions.

We appreciate the reviewer for pointing this out and agree with the reviewer’s interpretation. Indeed, according to our data, the importance of TE-mediated epigenetic effects in shaping genomic TE abundance mainly acts on a short evolutionary time scale.

Previous studies have found greater net gains of intron sequence and duplicated genes in *D. yakuba* than in *D. melanogaster* and/or *D. simulans*. These observations suggest that the *D. yakuba* genome may be more tolerant of gained sequence. If similar mechanisms apply to the other two species in the *yakuba* species complex, lineage-specific mechanisms that shape the evolution of the whole-genome may also play a vital role in determining genomic TE abundance, and might potentially override the influence of selection against TE-mediated epigenetic effects. We revised our conclusion (by emphasizing it supports association within species complex, p30, line 8-12 (in Results)) and included these discussions in and p41, line 16-29 (in Discussions).

8. Biological difference between H3K9me2 magnitude and spreading extent seems unclear. Roughly speaking, the beginning of the manuscript made the two seem similar, the middle part found an effect on the extent only, and the last part found an effect on the magnitude only.While I appreciate the authors examining two aspects of H3K9me2 spreading, I am unsure what either aspect truly reflects. I worry that some readers would find the authors cherry-picking either aspect to support their claims – perhaps authors should've commented a priori the difference between the two aspects such that conclusions made on one, but not the other, aspect of TE-mediated epigenetic effects are more convincing.

We agree with the reviewer that we did not make clear distinctions between the magnitude and extent of H3K9me2 enrichment, which could confuse readers about why we would need both indexes.

Previous studies on the spreading of repressive marks from constitutive heterochromatin suggested that the magnitude and extent of H3K9me2 enrichment may not be perfectly associated, mainly because the local genomic context could influence the extent of the spread. If similar molecular mechanisms are also applicable to epigenetically silenced TEs in euchromatin, the magnitude and extent should provide different information about TE-mediated local enrichment of repressive marks. Before conducting this study, we had no a priori expectation for which index would be more relevant to the questions we aim to address. Accordingly, we feel it is most appropriate to present both of them. Especially, in several analyses where the results differ between the two indexes, they provide important information about the underlying mechanisms (see below).

According to our observations, the major difference between these two indexes is that the magnitude of TE-mediated H3K9me2 enrichment is more significantly associated with lower TE population frequencies (i.e., strong evidence supporting being selected against) and is correlated with genomic TE abundance. On the other hand, the impact of neighboring gene expression only influences the extent, but not the magnitude, of TE-mediated H3K9me2 enrichment. In the discussion, we interpreted these two findings together as the result of the extent being more sensitive to the local genomic context (which is similar to previous studies) than the magnitude, making the latter a more direct estimate for the functional and thus fitness impact (which we still don’t know, see p40, line 18-p41, line 2 for discussion) of TE-mediated H3K9me2 enrichment.

To make better distinctions between these two indexes, we (1) included definitions for what these two indexes are at where they were first mentioned in Results (instead of just in Materials and methods; p10, line 3-9), (2) discussed previous observations to explain why we used both indexes and provided our a priori guess that the extent may be more sensitive to local genomic context (p10, line 19-p11, line 8), (3) moved discussions regarding the difference and relative importance of these two indexes from the end of Discussion to the front of Discussion (p38, line 23-p39, line 7), and (4) emphasize their difference throughout the text when necessary. We hope these changes addressed the reviewer’s concerns.

9.1. Chromatin landscape was examined by Su(var) gene expression and satellite repeat abundance, which seems insufficient and incomplete. Chromatin landscape is too broad of a term when authors only examined Su(var) gene expression and simple repeat abundance. Have authors look at other heterochromatin genes (e.g., ZAD-ZNF, piRNA pathway) besides Su(var)?

(We split the reviewer’s comment #9 into two and answered them individually). Most of the piRNA pathway genes are expressed in the female germline and at the early embryonic stage. Accordingly, nearly none of them are expressed in our dataset, which was generated using 16-18hr late-stage embryos. In *Drosophila,* ZAD-ZNF proteins have a wide range of functions. For a ZAD-ZNF gene that has a demonstrated role in heterochromatin function (Oddjob, a Su(var)), we included it in our analysis.

It is worth noting that, instead of looking at the initiation of the epigenetic silencing of TEs, we mainly focus on the inadvertent side effect associated with the silencing – the enrichment of repressive marks at flanking non-TE sequences. This phenomenon is similar to the well-studied Position Effect Variegation (PEV). Accordingly, our analysis focuses on Su(var), whose role in determining the strength of PEV has been well characterized. To explain this point, we included more motivation for looking at Su(var) at p31, line 8-15. We agree with the reviewer and tone down several places that our study mainly concerns repressive chromatin landscape (e.g., p38, line 1) in addition to other existing places.

9.2 What is the relationship between simple repeat abundance, TE abundance, and genomewide repeat content? This latter point is related to the point #6 above about euchromatic TE vs. genome-wide TE. I kept wondering, in the context of this work, whether there is biological difference between satDNA and TE – are they similar because they cause similar epigenetic effects or do they differ?

As described in our response to reviewer’s comment #6, our study mainly concerns the evolutionary dynamics of TEs in the euchromatic genome. We hope changes made while addressing that comment clarify this point.

We are unaware of studies that report H3K9me2/3 enrichment mediated by satellites in the *Drosophila* euchromatic genomes, except for those extreme cases through experimental manipulation (e.g., a large tandem array of constructs). The replication/expansion mechanisms and potential fitness impacts also differ between TEs and satellite repeats. Accordingly, we would argue the evolutionary dynamics would differ between these two types of repetitive sequences.

Minor comments:– Figure 2 beautifully shows the epigenetic effect depends on the TE family and other factors. I wonder if the authors would want to expand Figures3-5 to see if any TE family exerts a particularly strong (or weak) epigenetic effect in different species?

Because there are only a few TE families that have at least five copies included in the analysis for all genomes, our power to perform the suggested analysis is limited. Also, in Figures 3-5, our main goal is to identify the associations between the strength of TE-mediated epigenetic effects and various factors (e.g., gene expression, genomic TE abundance), irrespective of TE identity.

Due to these two reasons, we did not perform the analysis suggested by the reviewer.

Nevertheless, we hope the data presented in Figure 2C provides a glimpse into the reviewer’s question. We added some more discussions in the text regarding this p12, line 21-26, pointing to TE families that are of particularly strong/weak epigenetic effects across species.

– Figure 6 cartoon probably should have K9 spread less towards the gene side.

We agree with the reviewer and revised the figure accordingly.

Reviewer #1 (Significance (Required)):The senior author has previously found that euchromatic TE exerts an epigenetic effect via H3K9me2 spreading, which confers a fitness cost. Now, this work builds on the previous finding to attempt a conceptual advance that such a TE-mediated epigenetic effect differs across six species examined and depends on species-specific chromatin landscape. The examination of six species and the connection between molecular biology, population genetics and potentially genome evolution are rare among existing literature and thus present novelty. It should interest people working on TE, heterochromatin, epigenetics and population genetics. My expertise is in chromatin, epigenetics and RNAi. I do not have sufficient expertise to evaluate the regression analysis and TE population frequency estimation done in this work.Reviewer #2 (Evidence, reproducibility and clarity (Required)):Summary: In this article, Huang and colleagues investigate the degree to which TEs can induce heterochromatin formation within the euchromatin across species. They hypothesize that different TE abundances across species might be explained by variation in the degree to which TEs impact a selective effects. In this article, they show that there does seem to be a big difference in the magnitude of H3K9me2 induction between the mel clade and the yak clade. This is extremely interesting and very worthy of publication. The authors do a great job exploring a variety of alternate hypotheses.Major Comments:I have no major comments that may significantly alter the conclusions. I found their approach robust. However, I will admit there is one thing I find difficult to understand regarding their model and conclusion. In the introduction they state:"These observations spurred our previous hypothesis that varying epigenetic effects of TEs could result in between-species differences in the strength of selection against TEs, eventually contributing to divergent genomic TE abundance (Lee and Karpen 2017)."And in the final figure (figure 6) they provide a schematic of this hypothesis (note typo in the worded "mediated" in the figure). In particular, they show (indicating the * where they state the paper found) that (1) The magnitude of Heterochromatin induction can lead to (2) lower TE population frequencies and this can lead to (3) lower genomic TE abundance. But doesn't 5A show that the yak clade has HIGHER magnitude of heterochromatin induction (as shown earlier) but also higher estimated TE copy number? So, 5A shows the opposite conclusion with respect to genomic copy number. I feel the disconnect might arise from mixing up the notions of low frequency of TE insertions from the notion of genomic copy number. It is possible to have a VERY large number of TEs that are segregating at very low numbers. So, selection against TEs (as evident by low allele frequency) doesn't easily translate to total copy number. I hope this makes sense.

We appreciate the reviewer for pointing this out and this is indeed something we should address. Part of the issue raised by the reviewer is hopefully addressed in our response to reviewer #1’s comment #7 (please see above). In short, the strong species complex effect suggests that the role of TE-mediated epigenetic effects in determining genomic TE abundance may be acting on a short evolutionary time scale and could be overridden by other general processes that determine gain/loss of sequences (irrespective of types). This may explain why TEs in yakuba complex species show stronger epigenetic effects but are also more abundant.

We agree with the reviewer that our model (presented in Figure 6) has unstated underlying assumptions. According to the population genetic theory of TE copy number, the equilibrium copy number is determined by the strength of selection removing TEs and TE increase through transposition rate. Our model assumes that the transposition rates are similar, and TE copy numbers are at equilibrium between species. Here, the population frequencies of TEs serve as an indicator for the strength of selection removing TEs. It is possible to have a large number of TEs segregating at low population frequencies, but the average copy number across individuals of a species (i.e., the number of TEs in each genome) would still be small. We hope this interpretation makes sense to the reviewer and are open to suggestions to further revise the model. We included additional discussions regarding the assumptions of our model, which can be found at p41, line 7-10 and in Figure 6 legend.

Minor Comments: I have some minor comments for clarity.1) Since the paper is motivated to explain differences between species in genomic copy number, this information should be provided in the first figure in 1A (alongside the species name, with the two sim and yak samples also both provided numerically)

We revised Figure 1A according to the reviewer’s suggestion.

2) Something should be made of the fact that some of these arguments can be circular and the arrow of causality is hard to tease apart. Genomic TE abundance may be explained by species level differences in the epigenetic impact of TEs. But, it may also be the case that as TE abundance varies, there may be evolution of the epigenetic impact. Causality is tricky here and worth being up front about.

We agree with the reviewer, and we included this possibility in Discussion. In this revised manuscript, we rewrote the discussion to further emphasize this point (p42, line 1-8).

3) Some clarity in the text (not just in the methods) should be provided for the reader about how the "extent of spreading" was calculated.

We agree with the reviewer and added these details in the Results section when this index was first mentioned (p10, line 3-9).

4) In figure S1, what does it mean for points to have zero magnitude of enrichment but a great extent of enrichment? Shouldn't all points for zero magnitude be at the origin? I am looking at the scattered points that run along the Y axis in S1 of just the mel group (not seen in yak group)

We greatly appreciate the reviewer for pointing this out.

We identified an error in the coding for generating this plot. In this script, a few “NA”s in the data file were treated as 0 when calculating the average magnitude of the two replicates. This error affected the results of Figure 1- Supplementary Figure S1, Figure 2B and 2C, Figure 2 – Supplementary Figures S1 & S2. We updated these figures and revised the correlation coefficients in Figure 1 – Supplementary Figure S1. After removing these TEs with erroneous estimates, the correlation between the magnitude and extent of TE-mediated epigenetic effects becomes stronger. Other updated results reached the same conclusions as previous results.

We thoroughly checked our coding, and all the other results throughout the study are unaffected. For other analyses, we used a different script in R, which removes the “NA”s.

While addressing this comment, we noticed that we forgot to provide details about why a TE may have NA estimates for TE-mediated epigenetic effects (low local sequencing coverage) and how we treated them in the analysis (remove them from the analysis). We included these details in p48, line 20-21; p48, line 31.

5. In S2, how they compare the correlations across two genomes is confusing. Is that for shared TEs? Also, do estimates from two strains (Y axis) include values from X axis? It seems like this is saying X is correlated with the average of X and Y, which will always be true if Y is a random variable. Overall, I was a fairly confused about what S2 was showing.

We agreed with the reviewer that neither the main text nor the legend for Figure1 –

Supplementary Figure 1 (previous Figure1 – Supplementary Figure 2) contains sufficient details for how these two estimates were generated.

The concepts for estimating the magnitude and extent of TE-mediated epigenetic effects are similar for using one genome and two genomes. The major difference is the baseline used. For one genome, we used H3K9me2 enrichment > 1 as a criterion to estimate the extent. In contrast, for two genomes, the comparison is made between homologous sequences of the focal genome with the focused TE insertion to the alternative genome without the TE insertion (i.e., whether H3K9me2 enrichment in strain 1 > H3K9me2 enrichment in strain 2). For the magnitude of TE-mediated H3K9me2 enrichment, the two-genome estimates were further normalized by the enrichment level in the homologous sequence without the TE insertion. We agreed with the reviewer that these two estimates are related due to shared variables, but the relationship between them is not a simple arithmetic calculation. Accordingly, we feel that presenting this correlation result is still important.

We included some more details in the main text (p10, line 9-13) and Figure 1 – Supplementary Figure S1 legend that hopefully could help readers interpret the result. We also noted that detailed methods can be found in Materials and methods.

When revising according to this comment, we noticed that we did not include method details about how we identify homologous regions in the two PacBio genome assemblies. We added these details (using minimap) in p50, line 22-25.

6) In figure 2A. Do not includes lines connecting values between species. Just used colored dots This is not a series. I think this issue is elsewhere as well.

We appreciate the reviewer for pointing out our oversight and revised Figures 2A and 2C accordingly.

Reviewer #2 (Significance (Required)):This article provides some extremely important insight into the manner in which heterochromatin induction can shape TE dynamics across species. What explains variation in genome size and TE content is one of the big questions in evolutionary genomics. The results are somewhat complex, but that is because biological systems are complex.My expertise is in evolutionary genomics of TEs. I am not an expert in chromatin biology and regulation of gene expression.Reviewer #3 (Evidence, reproducibility and clarity (Required)):In this manuscript, Huang et al. expand upon results presented in two previous influential publications led by the corresponding author of this manuscript: Lee (2015) and Lee and Karpen (2017). Lee (2015) reported that H3K9me3 spreads from euchromatic TE insertions, these insertions lead to reduced expression of nearby genes, and this phenomenon depends on the piRNA pathway. Lee and Karpen (2017) reported a significant negative correlation between both H3K9me2 magnitude and extent and TE population frequencies in *D. melanogaster* and demonstrated strong spreading of H3K9me2 from TE insertions, an average of 4.5 kb into flanking regions. They also report stronger epigenetic effects of TEs in *D. simulans* compared to *D. melanogaster* which they propose could be due to higher expression of Su(var) 3-9.Here, Huang et al. examine H3K9me2 magnitude and extent (i.e. spreading) across six species of *Drosophila*, including two different strains of D. simulans and D. yakuba. Across all species and strains, the authors observe a strong enrichment of H3K9me2 at euchromatic TE insertions. They find a large amount of variation in H3K9me2 magnitude and extent across TE classes and families, and among species and strains, but this variation does not appear to show any meaningful and/or consistent pattern. While they find that genic H3K9me2 is positively associated with the magnitude and extent of TE-derived H3K9me2, only the magnitude effect depends on gene distance. They find no relationship between TE epigenetic effects and expression of adjacent genes and some evidence of asymmetric H3K9me2 spreading due to neighboring highly expressed genes. Across all species, the magnitude of H3K9me2, but not the extent, was negatively correlated with TE population frequency. Similarly, the magnitude of H3K9me2, but not the extent, was correlated with TE abundance across species. Finally, they find that the magnitude of H3K9me2 (extent not tested) is positively correlated with expression of Su(var) genes across species.The analyses performed here are rigorous and the appropriate replicates have been included. The authors also clearly go to great lengths to try to address confounding factors that could explain the correlations they are seeing.1. My main concern related to evidence and reproducibility is that the results shown for H3K9me2 magnitude and spreading are highly variable between species and strains. It is not clear to me how much of this variation is due to biology versus technical issues related to ChIP, however I am surprised at how low the correlations among replicates are for these measures (Figure 1, Supplementary Figure S4). I would encourage the authors to perform additional QC measures for their ChIP experiments and replace any low quality replicates so as to minimize the contribution of technical variation. Some QC suggestions: the K-MetStat Panel spike-in should allow for an assessment of antibody specificity for the K9me2 modification and the Irreproducibility Discovery Rate (IDR) framework can quantify reproducibility among replicates.

We appreciate the reviewer’s helpful comment and performed analyses following the reviewer’s suggestion. We used ChIP-seq reads mapping to SNAP-ChIP K-MetStat Panel to assay the specificity of the used antibody (Abcam 1220). Consistent with previous investigations by modEncode (Egelhofer et al. 2011) and others (Sentmanat and Elgin 2012), our analysis found that this antibody shows high specificity against H3K9me2. We included this result in p45, line 23-25 and Figure 1 – Supplementary Figure S5.

We performed the suggested IDR analysis and found no direct associations between IDR and the correlations between replicates for the index of TE-mediated epigenetic effects (the diagnostic IDR plots are now in Figure 1 – Supplementary Figure S7). For instance, *D. melanogaster* shows the lowest correlation for the magnitude of TE-mediated H3K9me2 enrichment between replicates, but has decent consistency between replicates by IDR analysis.

A plausible technical reason for this discrepancy is that IDR analysis relies on called peaks. Custom pipelines, which were designed to identify “sharp peaks,” are likely to miss the enrichment of repressive marks, which typically have broad and lower level of enrichment (reviewed in Park 2009 Nature Review Genetics). Accordingly, TE-mediated H3K9me2 enrichment in euchromatic genome could be missed in the IDR analysis. Indeed, a previous study found that only 12% of TE-induced H3K9me2 enrichment was detected by peak calling (Lee et al. 2020 PLoS Genetics). Another more important biological reason is that the spreading of repressive marks from constitutive heterochromatin has been observed to be a variegating phenotype and could differ between cells and individuals. Accordingly, the variability for the magnitude and extent of TE-mediated local enrichment of heterochromatic marks could have contributed to the low correlation between replicates for some samples. We included these discussions in p49, line 10-29.

Thanks to the reviewer’s comment, we noticed that *D. simulans* strain 2 has much fewer peaks showing low IDR between replicates. To be conservative, we thus tried excluding this sample for the key analysis for our study and still found negative associations between the magnitude of TE-mediated H3K9me2 and genomic TE abundance. We included this analysis in p29, line 2224; p29, line 28-29; p49, line 29-p50, line 3.

2.1. Related to the comment above: does the "extent" measurement used here differ from the "spreading" measurement used in Lee (2015) and Lee and Karpen (2017). If so, why was a new metric used?

(We split the reviewer’s comment #2 into two and answered them individually).

We are a bit unsure about the “spreading” measurement the reviewer is referring to. In Lee (2015), we did not estimate TE-mediated local enrichment of H3K9me3 and instead investigated the associations between genic H3K9me3 enrichment and the presence of adjacent TEs. In Lee and Karpen (2017), we used the same measurement as the current study – the extent and magnitude of TE-mediated local enrichment of H3K9me2.

We agreed with the reviewer that we did not include enough descriptions of the two indexes for the epigenetic effects of individual TE and this can cause significant confusion. To address this, we added text in p10, line 3-9.

2.2. Could that explain why spreading no longer seems to be an important contributor to TE epigenetic effects?

We agree with the reviewer that we did not clearly define “epigenetic effects of TEs” and, without that, the term may implicitly imply functional consequences (e.g., gene expression). We used the term “epigenetic effects” to describe TE-mediated local enrichment of H3K9me2. We added definition for this term when first mentioning this term in Introduction (p5, line 28) and Results (p10, line 3-5).

The spreading of H3K9me2 beyond TE boundaries could lead to local enrichment of H3K9me2. The strength of such an event could be described by the enrichment level (magnitude) and extent (extent) of the local enrichment of H3K9me2 outside TE sequence. We see that the word “spreading” might also be interpreted as the “extent” of the H3K9me2 enrichment. Accordingly, we replaced this word throughout the manuscript with a more accurate description (“local enrichment of H3K9me2”) to avoid confusion and misinterpretation (e.g., p1, line 23; p4, line 15; p5, line 1; p38, line 10).

According to Figure 1B and C, we observed that the presence of TEs not only lead to H3K9me2 enrichment at immediate TE-flanking regions, but also could extend for several kbs, suggesting that the “spreading” (used in reviewer’s context) of repressive marks from TEs is still an important “epigenetic effects.” We also observed that the extent of TE-mediated enrichment of H3K9me2 is positively correlated with genic H3K9me2 enrichment (Figure 3A). It is when we investigated the role of TE-mediated local enrichment of H3K9me2 that we did not find the predicted associations between the *extent* of such effect and genomic TE abundance, but did find significant associations between the *magnitude* of such effect and TE abundance. It is worth noting that Lee and Karpen 2017 made similar observations with two species (i.e., TE copy number is associated with the magnitude of epigenetic effects, but not with the extent), which we noted in p30, line 1-3. In addressing the reviewer #1’s comment 8, we also made more clear distinctions between the two estimates (magnitude and extent of TE-mediated H3K9me2 enrichment; see above).

3. In terms of clarity: if the authors demonstrate that their ChIP replicates are highly reproducible and the "extent" metric is not meaningfully different from the previously used "spreading" metric, then it is hard for me to think of technical or analytical reasons why the findings reported here differ substantially from those reported previously. The previous model was intuitive: an underappreciated aspect of the deleterious nature of TE insertions is that euchromatic TEs can initiate heterochromatin formation which spreads into nearby genes leading to harmful reductions in their expression. This study suggests that magnitude of H3K9me2 is the major contributor to the "epigenetic effect" of TEs and that there is no relationship between TE-induced H3K9me2 and neighboring gene expression. I would encourage the authors to use the discussion to acknowledge these contradictions and try to reconcile these results with those from the previous two papers. Is there reason to believe the previous studies were incorrect? Otherwise, it is very hard to know what to take away from this study.

We agreed with the reviewer that without clearly defining “epigenetic effects,” readers could get the impression that the current study contradicts previous findings. Hopefully, our revision according to the reviewer’s comment #2 above would clarify the meaning of “epigenetic effects of TEs.” Also, as noted above, similar to Lee and Karpen (2017), we used two indexes to describe the epigenetic effects of TEs, or TE-mediated local enrichment of H3K9me2 in the euchromatic genome. Both the previous study with a limited number of species and our study found that (1) TEs lead to a significant increase in the magnitude and extent of local enrichment of H3K9me2, and (2) TE copy number is negatively associated with the magnitude, but not extent, of the effect. Our findings do not contradict previous studies.

The generally limited associations between TE-mediated epigenetic effects and lowered gene expression, as mentioned by the reviewer, was also reported in other model species, including in *D. melanogaster* (Lee and Karpen 2017 *eLife*), *Arabidopsis thaliana* (Quadrana et al. 2016 *eLife*; Stuart et al. 2016 *eLife*), rice (Choi and Purugganan 2018 MBE), and mouse (Pezic et al. 2014 Genes & Development). This has been a perplexing finding in the field of TE evolution, which was extensively discussed in two recent reviews (Kelleher et al. 2020 Trends in Genetics; Choi & Lee 2020, PLoS Genetics; discussed in our Introduction p5, line 9-16). Our current study investigated this question in multiple species and used both within-genome and across-genome (i.e. between homologous alleles) comparisons, further revealing the complex relationship between TE-mediated epigenetic effects and neighboring gene expression. We hope that our new analyses addressing reviewers' comments (reviewer #1’s comment 3 and reviewer #3’s comment 4) further help corroborate our findings. We also included additional discussions regarding the limited associations between TE-mediated epigenetic effects and gene expression (p39, line 9-22) and provided possible reasons why we did not find the expected negative associations between them (p39, line 22 – p40, line 16).

It is worth noting that our analysis focuses on identifying the genome-wide average pattern; it is still plausible that the local enrichment of H3K9me2 induced by individual TE occasionally lowers nearby gene expression (e.g., genes with positive z-scores in Figure 3C). In fact, many early examples of TE-mediated enrichment of repressive marks were discovered by the phenotypic consequences of reduced neighboring gene expression (reviewed in (Choi and Lee 2020 PLoS Genetics)). We included this discussion in p18, line 14-21.

In this study, we tested the model proposed by Lee and Karpen (2017) (Figure 6). We were able to find support for most of the models except for the generally expected, but unsupported prediction that TEs’ epigenetic effect would lower neighboring gene expression. We proposed several possibilities for this missing link, which can be found in p40, line 18 – p41, line 2.

Other major comments:4. Figure 3C: For the gene expression analysis, it would be helpful to see a scatterplot where each point is a gene and the axes are the log2 expression level of the gene in the strain with an adjacent TE versus the log2 expression of the same gene in the strain without an adjacent TE. The slope of the regression line would show whether there is any tendency for genes with adjacent TEs to have lower expression.

We performed the suggested analysis following the reviewer’s suggestion (p18, line 9-11 and Figure 3 – Supplementary Figure 9). In these plots, we have the log2 expression level of genes in the strain with a TE on the X-axis and in the strain without TE on the Y-axis. We plotted genes whose nearest TEs with and without epigenetic effects with different colors. The slope is close to 1 (i.e., no difference in gene expression with and without TE presence), and there is no difference in slope for genes whose nearest TEs with and without epigenetic effects. These observations are consistent with our z-score analysis, finding a limited influence of TE-mediated epigenetic effects on gene expression.

5. It is interesting that there is clear evidence supporting enrichment of H3K9me2 at TEs leading to genic H3K9me2 enrichment but not expression downregulation. It would be very interesting if the authors could extend this analysis to include all genes from the euchromatic chromosome arms showing H3K9me2 enrichment in at least one strain or species, irrespective of TEs. Overall, are differences in genic H3K9me2 enrichment between strains/species associated with differences in gene expression?

This is a very interesting question raised by the reviewer. Yet, addressing this question is beyond the scope of this study. The suggested analysis would address the general role of the enrichment of H3K9me2 in gene expression, which is a broad topic.

It may be worth noting that, according to a previous analysis (Lee et al. 2020 PLoS Genetics), most, if not all, of the polymorphic H3K9me2 enrichment in the euchromatic genome is due to the presence/absence of TEs within species. Few H3K9me2 enriched regions in the euchromatic genome will thus be TE-free and could be suitable for performing the suggested analysis. On the other hand, the between-species comparison of gene expression is likely to be confounded by factors other than the enrichment of H3K9me2 (e.g., the evolution of regulatory sequences) and thus be challenging to interpret. We thus did not proceed with the suggested analysis.

6. I'm confused by the difference between Figure 3D and Figure 3 – Supplementary Figure S7. If I understand correctly, for *D. melanogaster*, there is no difference in the extent of spreading between the two sides of a TE when the highly expressed gene is near the TE, nor is there a difference when the highly expressed gene is far from the TE (Figure 3D). However, when expression measurements are used from a different strain (Figure 3 – S7), there is a clear difference in the extent of spreading on the gene side, specifically when the highly expressed gene is close to the TE. What explains these contradictory results? Are a large proportion of the genes identified as highly expressed when using the other strain's data no longer highly expressed in the strain of interest? (presumably due to the effect of the nearby TE?). If so, that seems notable as it lends support to an association between spreading and gene downregulation.

We appreciate the reviewer for pointing out this incongruence. Following the reviewer’s suggestion, we counted the number of genes that have a TE nearby and are categorized as highly expressed-

In both our data and modEncode: 179

Only in the modEncode data: 93

Only in our data: 83

There is no significant difference in the number of genes that are categorized as highly expressed in only modEncode data or only in our data. Accordingly, the suggested possibility (the difference between Figure 3D and previous Figure 3 – Supplementary Figure S7 is due to TE’s epigenetic effects lowering gene expression) may not be applicable to our observations.

The main reason why the result in Figure 3D differs from those of previous Figure 3 –

Supplementary Figure S7 is probably due to the fact that we used different statistics. In Figure 3D, we estimated the z-score for the difference in the extent of TE-mediated H3K9me2 enrichment between the genic and intergenic sides, normalized by the standard deviation of the estimates. In previous Figure 3 – Supplementary Figure S7, we simply compared the extent of TE-mediated H3Kme2 enrichment on the two sides of a TE. The z-score should be a more appropriate statistic given that it is normalized, so comparable across TE insertions. By using the z-score to perform the analysis in previous Figure 3 – Supplementary Figure S7, we found the same results like those in Figure 3D, and there is no statistically significant difference in the extent of TE-mediated H3K9me2 enrichment between the two sides of a TE in *D. melanogaster*.

**Author response image 1. sa2fig1:** 

Given that there is no major difference between Figure 3D and updated Figure 3 – Supplementary Figure S7, and the latter does not add additional information, we decided to exclude analysis based on categorizing genes according to modEncode data (i.e., previous Figure 3 – Supplementary Figure S7).

7. The authors report that H3K9me2 spreading from TEs is counteracted by the presence of a highly expressed gene nearby (either upstream or downstream), leading to asymmetric spreading. However, the spreading extent index is calculated by averaging the extent of spreading between the two flanking regions of the TE. Is this the best choice given the presence of asymmetric spreading? Would using the longer of the two distances be better?

We think the reviewer is suggesting we use the “intergenic side,” which is less likely influenced by the expression of adjacent genes. Our response is based on this interpretation.

According to our data presented in Figure 3D, the extent of TE-mediated H3K9me2 enrichment is not symmetrical even for TEs far from genes. This is consistent with findings from constitutive heterochromatin that the spreading of repressive marks is a stochastic process. Also, our goal is to understand the potential functional and thus fitness consequences of TE-mediated H3K9me2 enrichment; such effect on the genic side may thus be of higher importance. Accordingly, it is challenging to predict a priori which side of the extent of H3K9m2 enrichment is more important, and we still chose to use the average of the two sides for our analysis.

8. An important prediction related to the proposed relationship between differences among species in expression of Su(Var)s and differences in the magnitude of H3K9me2 at TEs is that the same pattern should be present for other heterochromatic loci. Do other (non-TE) heterochromatic loci show stronger H3K9me2 enrichment in species with higher expression of Su(Var)s?

We agree with the reviewer that this is an interesting question. Yet, the suggested analysis and interpretation of the results are outside the scope of the current study. In addition, the effects of reduced dosage effects of Su(var)s were mainly observed at the boundaries of heterochromatin (e.g., at translocated genes located at the heterochromatin/euchromatin boundary and associated with PEV phenotype, reviewed in Elgin and Reuter 2013). In order to perform such analysis, one needs to first define the heterochromatin (HC)/euchromatin (EU) boundaries epigenetically using H3K9me2 enrichment. A potential issue is that the enrichment level of H3K9me2 at genes at HC/EU boundaries would highly depend on how the boundaries were first defined by H3K9me2, making the interpretation of the analysis challenging. Because of these reasons, we did not proceed with the suggested analysis.

9. It is not clear to me how the 1360 reporter experiment is applicable to the Su(Var)s expression results. The authors only consider H3K9me2 magnitude in their comparison to Su(Var)s expression rank yet, if I understand correctly, the 1360 reporter experiment measures spreading/extent of H3K9me2 which leads to downregulation of the adjacent mini-white gene. The reporter experiment described here suggests that reducing Su(Var) gene dosage leads to a reduction of H3K9me2 spreading along with a concomitant reduction in reporter gene silencing, however the other results reported here suggest that spreading of H3K9me2 is not an important contributor to the epigenetic effects of TE nor does spreading lead to gene downregulation.

We agree with the reviewer that our motivation for performing the 1360 reporter experiment was unclear and needs to be further clarified. As discussed in response to the reviewer’s comments 2 & 3, we did find that euchromatic TEs lead to local enrichment of H3K9me2, which is characterized by higher H3K9me2 enrichment level (magnitude) and an extent beyond the TE sequence. Even though on a genome-wide average, we did not find predominantly negative impacts of TE’s epigenetic effects on neighboring gene expression, our analysis does not preclude individual cases where a gene’s expression is reduced by TE-mediated enrichment of H3K9me2 (see discussions in comment 3 above and p18, line 14-21).

The 1360 reporter consists of a TE right next to the reporter gene. Accordingly, for this reporter system, the magnitude, instead of the extent, of TE-mediated H3K9me2 enrichment may be more important. In this specific reporter system, there are associations between the presence of 1360, the enrichment level of H3K9me2 at the mini-white reporter gene, and the reduced amounts of red-eye pigmentation (likely due to reduced gene expression). Accordingly, the eye pigmentation level in this system serves as a convenient readout for the magnitude of TE-mediated H3K9me2 enrichment. We clarified these points in p33, line 23-29.

Minor Comments1. Lee (2015) examined H3K9me3 while Lee and Karpen (2017) and this manuscript study H3K9me2 levels. I think it is important to explain why one modification was used versus the other. Do you expect them to be redundant or might they play different roles in spreading versus magnitude of heterochromatin?

We agree with the reviewer that we did not have enough context for why we chose to study H3K9me2. Both H3K9me2 and H3K9me3 are found to be highly enriched in the heterochromatic regions of the genome in *D. melanogaster* (Riddle et al. 2011 Genome Research; Kharchenko et al. 2011 Nature). According to Lee and Karpen (2017), both H3K9me2 and H3K9me3 captured TE-mediated local enrichment of repressive epigenetic marks. The major reason that our study focused on H3K9me2 is due to the availability of suitable antibodies. Antibody against H3K9me2 is monoclonal and has been validated by modEncode to show high consistency (Egelhofer et al. 2011), while commonly used H3K9me3 ChIP antibody available at the time when this study started (Abcam ab8898) is polyclonal and likely cross-react with other histone modifications, in particular H3K27me3 (personal communications with modEncode team). Because H3K9me2 and H3K9me3 would both provide the needed information about TE-mediated local enrichment of heterochromatic marks, we proceeded with only H3K9me2. To provide more context, we added text at p7, line 6-12.

2. For the TE population frequency analysis, I was not able to find the cutoffs for what were considered high frequency and low frequency TEs

This is an error at our end; we forgot to include details about the categorization of TEs based on population frequencies. High-frequency TEs are those that are found in both the studied strain and the population sample, while low-frequency TEs are those that are unique to the studied strain and absent in the population sample. We included this information p23, line 19-21.

3. The Su(Var) section is a bit confusing to interpret because gene rank should correlate negatively with H3K9me2 but expression level should correlate positively and thus, both positive and negative correlations are referenced in this section when describing the same pattern.Could you switch the gene rank order (i.e. lowest to highest) so that the expected correlation is in the same direction regardless of whether you are referring to expression level or expression rank?

In order to be consistent, we used expression rank for all expression analyses presented in Figure 3 (TE’s impact on gene expression) and Figure 5 (Su(var) expression). To help readers interpret the results, we added additional texts to either define expression rank (“rank from the highest expressed genes”) or interpret the result (“lower expression rank (i.e., higher expression)”, e.g., p16, line 7; p17, line 26). We hope these could help readers and avoid confusion.

4. Is there a relationship between local recombination rate and H3K9me2 extent and/or magnitude? If so, that could also be a potential confounder with respect to TE population frequency. I believe this was addressed in Lee and Karpen (2017) but it would be useful to reiterate here.

We repeated a similar analysis as Lee and Karpen (2017) in *D. melanogaster,* the only species in which a high-resolution recombination map is publicly available. Consistently, we found no associations between TE-mediated epigenetic effects and local recombination rate, further corroborating the previous analysis. We included these results in p24, line 17-29.

Reviewer #3 (Significance (Required)):This study builds upon two previous studies that have made a large impact on the field. This work is significant because it suggests that the epigenetic effects of euchromatic TE insertions are more complicated than previously suggested by work that was only conducted in D.melanogaster. It also suggests that the mechanisms by which TEs exert these effects and the mechanisms that make these effects harmful need to be revised and studied further.This work will be of general interest to those whose work involved evolutionary biology, population genetics, epigenetics and chromatin, and transposable elements.My expertise is in the evolutionary and functional genomics of *Drosophila*.lex and 3 in yakuba species comple